# Peptidoglycan synthesis drives a single population of septal cell wall synthases during division in *Bacillus subtilis*

Kevin D. Whitley [1] ✉, James Grimshaw[1], David M. Roberts[2], Eleni Karinou[1], Phillip J. Stansfeld [2,3] & Séamus Holden [2] ✉

Bacterial cell division requires septal peptidoglycan (sPG) synthesis by the divisome complex. Treadmilling of the essential tubulin homologue FtsZ has been implicated in septal constriction, though its precise role remains unclear. Here we used live-cell single-molecule imaging of the divisome transpeptidase PBP2B to investigate sPG synthesis dynamics in *Bacillus subtilis*. In contrast to previous models, we observed a single population of processively moving PBP2B molecules whose motion is driven by peptidoglycan synthesis and is not associated with FtsZ treadmilling. However, despite the asynchronous motions of PBP2B and FtsZ, a partial dependence of PBP2B processivity on FtsZ treadmilling was observed. Additionally, through single-molecule counting experiments we provide evidence that the divisome synthesis complex is multimeric. Our results support a model for *B. subtilis* division where a multimeric synthesis complex follows a single track dependent on sPG synthesis whose activity and dynamics are asynchronous with FtsZ treadmilling.

Cell division is a basic requirement for bacterial life and a major antibiotic target. At a molecular level, division is a remarkable feat of engineering by the bacterial cell: a set of nanoscale proteins must cooperate over large distances to build a micron-scale cross-wall (septum) across the middle of the cell against heavy outward pressure. Many of the proteins involved are highly conserved across the bacterial domain, including the septal peptidoglycan (sPG) synthases that insert new cell wall material into the ingrowing septum and the cytoskeletal tubulin homologue FtsZ[1]. FtsZ forms short filaments that move around the division septum by treadmilling[2–5]—a type of motion where monomers bind one end of a filament at the same rate they unbind the opposite end, causing the filament to move forward even though monomers remain stationary. In contrast, sPG synthases are part of a larger divisome synthesis complex[6] that does not treadmill, but instead moves processively around the division ring[2,3,5,7].

Several models have been proposed to explain how these proteins cooperate to enact division despite such different motion patterns[8]. We previously proposed a model where FtsZ treadmilling drives septal constriction in *Bacillus subtilis* as a coupled cytoskeleton–synthase complex[2]. However, subsequent work in this and other Bacillota (also known as Firmicute) species demonstrated that sPG synthesis is not tightly coupled to FtsZ treadmilling. FtsZ treadmilling is dispensable for septal constriction after constriction has initiated in both *Staphylococcus aureus* and *B. subtilis*[4,9], and the motions of divisome synthesis complexes are uncoupled from treadmilling FtsZ filaments in *Streptococcus pneumoniae*[5]. Meanwhile, work on the Pseudomonadota (also known as Proteobacteria) species *Escherichia coli* and *Caulobacter crescentus* has supported a model where the synthase complex moves on two 'tracks': an FtsZ track where inactive synthase complexes are distributed around the division septum and an sPG track where synthase complexes build the cell wall independently of FtsZ[10–13]. According to

[1]Centre for Bacterial Cell Biology, Biosciences Institute, Faculty of Medical Sciences, Newcastle University, Newcastle upon Tyne, UK. [2]School of Life Sciences, University of Warwick, Coventry, UK. [3]Department of Chemistry, University of Warwick, Coventry, UK. ✉e-mail: kevin.whitley@newcastle.ac.uk; seamus.holden@warwick.ac.uk

this model, an activating protein (FtsN in *E. coli* and FzlA in *C. crescentus*) is required to initiate active sPG synthesis on the sPG track[13,14]. However, it is unclear how far this model generalizes across the bacterial domain, as many species lack a known activator of cell division.

In this Article, we investigated the dynamics of the divisome synthesis complex and its coordination with FtsZ in *B. subtilis*. We imaged the divisome transpeptidase PBP2B at a single-molecule level around the entire division septum by orienting cells vertically in bacteria-shaped holes[9,15]. In contrast to the predictions of the coupled-complex and two-track models, we found a single population of processively moving PBP2B that is dependent on sPG synthesis and not associated with FtsZ treadmilling. Although the motions of PBP2B and FtsZ are asynchronous, we found that the speeds of processive PBP2B molecules are partially dependent on FtsZ treadmilling. Additionally, we provide evidence that the divisome synthesis complex is multimeric. Our results support a model for division in *B. subtilis* where a multimeric divisome synthesis complex follows a single track dependent on sPG synthesis whose activity and dynamics are asynchronous with FtsZ treadmilling.

## Results

### Processive motion of the divisome requires sPG synthesis

We created a model of the *B. subtilis* divisome core complex (consisting of the proteins PBP2B, FtsW, FtsL, DivIB and DivIC) using AlphaFold2 Multimer (Methods and Extended Data Fig. 1), showing close agreement with the recent cryogenic electron microscopy structure of homologous proteins from *Pseudomonas aeruginosa*[6]. Since a recent study showed that these five proteins move together as a complex in vivo[7], we decided to follow the overall motion dynamics of the divisome synthesis complex by tracking the well-characterized transpeptidase PBP2B.

We constructed a strain that expresses a previously characterized HaloTag (HT) fusion of PBP2B as a functional sole copy at its native locus from an isopropyl β-D-1-thiogalactopyranoside (IPTG)-inducible promoter[2] (Methods and Supplementary Table 1). Induction of protein expression with 100 μM IPTG gave near-native cell morphology (Supplementary Fig. 1), with HT–PBP2B levels at ~67% of native PBP2B levels (Supplementary Fig. 2). We chose 100 μM IPTG induction of HT–PBP2B for experiments as higher induction levels did not produce substantial changes in cell morphology (Supplementary Fig. 1). To identify division septa, in addition to unlabelled native FtsZ our strain expresses green fluorescent protein (GFP)–FtsZ from an ectopic locus at low levels (0.075% xylose induction; Methods) that do not interfere with cellular growth (Supplementary Fig. 3) or FtsZ treadmilling speed (Supplementary Fig. 4).

To image single-molecule dynamics of PBP2B around the septum throughout division, we used smVerCINI (single-molecule vertical cell imaging by nanostructured immobilization), a method we developed for single-molecule imaging in rod-shaped cells by confining cells vertically in nanofabricated micro-holes[16] (Fig. 1a). For single-molecule resolution, we sparsely labelled cells with JFX554 HT ligand[17] (100–250 pM; Methods) and loaded them into agarose micro-holes as described previously[9,15]. We identified division rings using the dilute GFP–FtsZ signal, and then imaged single-molecule dynamics of HT–PBP2B in rich media at 30 °C (Methods, Fig. 1b,c and Supplementary Videos 1 and 2).

HT–PBP2B molecules showed several distinct motion behaviours. A distribution of linear track segment speeds reveals three populations: an immobile population (~0 nm s⁻¹), a processive population (~13 nm s⁻¹) (Fig. 1f and Supplementary Table 2) and a broad fast-moving population (>100 nm s⁻¹) that becomes visible when speeds are plotted on a logarithmic scale (Fig. 1f, inset). Individual HT–PBP2B molecules were also capable of transitioning between states (Extended Data Fig. 2). The probabilities of transitioning between states and their associated rates for all conditions in this study were calculated according to the formula described in Methods and are listed in

Supplementary Table 3. Based on the measured lifetimes of these states, an HT–PBP2B molecule under our standard conditions exists in the immobile state 38.1 ± 0.4% of the time, the processive state 59.0 ± 0.6% and the fast-moving state 3.0 ± 0.1% (mean ± standard error of the mean (s.e.m.); Supplementary Table 4).

We considered that the immobile population could represent molecules bound to the middle of treadmilling FtsZ filaments, as suggested previously for FtsW in *E. coli*[11]. However, the average lifetime of the immobile state of HT–PBP2B molecules (48 ± 3 s (mean ± s.e.m.); Supplementary Fig. 5) is substantially longer than the reported lifetime of FtsZ monomers at the division septum in *B. subtilis* (8.1 s)[7], making this situation unlikely. Due to the long lifetime of this population, we speculate that the immobile population may be bound to the cell wall. It is unclear how such binding would occur, although one attractive possibility is that immobile PBP2B molecules are bound non-covalently to acceptor peptides in the cell wall awaiting the emergence of nascent glycan strands to crosslink, as proposed recently for the *E. coli* elongasome transpeptidase PBP2 (ref. [18]).

Processively moving molecules showed a variety of noteworthy behaviours. Individual processive runs were usually terminal (54 ± 4%; mean ± s.e.m.; Supplementary Table 3), indicating that they ended in dissociation, photobleaching or the end of the acquisition period. However, they often ended with a change in direction (29 ± 3%), or sometimes by becoming immobile (12 ± 2%) or changing to the fast-motion state (5 ± 1%; Extended Data Fig. 2). We also observed numerous cases of processive tracks apparently crossing one another (Extended Data Fig. 3), suggesting that multiple divisome synthesis complexes exist in different lanes at the septal leading edge, slightly out of plane from one another. The speeds of processive molecules were independent of septal diameter (Supplementary Fig. 6), suggesting that their dynamics remain consistent throughout active constriction.

We wondered if the synthesis complex requires sPG synthesis for processive motion. To test this, we imaged the motion of HT–PBP2B immediately after treating cells with an excess (20 μg ml⁻¹) of penicillin G, which directly binds the enzyme's catalytic site and prevents transpeptidation. This abolished the processive population (13.4 nm s⁻¹ mean; 95% confidence interval (CI) 12.7–14.1), leaving only immobile and diffusive tracks (Fig. 1d,g and Supplementary Videos 3 and 4). We repeated the experiment with fosfomycin, a separate class of antibiotic that inhibits the synthesis of lipid II, the substrate for cell wall synthesis[19]. As with penicillin G, immediately after treatment with an excess (500 μg ml⁻¹) of fosfomycin, the processive population of HT–PBP2B vanished (Fig. 1e,h and Supplementary Videos 5 and 6). We conclude that processive motion of the synthesis complex requires sPG synthesis.

We next wondered what the broad high-speed population was. This population resulted from short back-and-forth tracks that appeared to show diffusive rather than processive motion (Extended Data Fig. 4a). These tracks were often located at larger radii than immobile or processive tracks (Extended Data Fig. 4b) and were nearly the only type of motion pattern outside the septal ring area (Supplementary Fig. 7), suggesting that they are not typically present at the septal leading edge and therefore unlikely to be involved in sPG synthesis. We were initially surprised to find an apparently diffusive population during our experiments, as we expected diffusive motion to be fast enough to be blurred out during our long (1 s) acquisition interval. Since linear speeds did not represent this population well, we instead measured the mean-squared displacements (MSDs) of these tracks and fitted an anomalous diffusion model to them (Methods and Extended Data Fig. 4c–f). The effective diffusion coefficient ($D_{eff} = 6.5 × 10^{-3} ± 0.4 × 10^{-3}$ μm² s⁻¹ (mean ± s.e.m.)) and diffusion exponent ($α = 0.80 ± 0.03$ (mean ± s.e.m.)) we obtained indicate that these tracks represent very slow subdiffusive behaviour. The low diffusion coefficient is similar to that previously measured for the *E. coli* elongasome transpeptidase PBP2 (ref. [18]), and suggests that diffusive HT–PBP2B molecules may

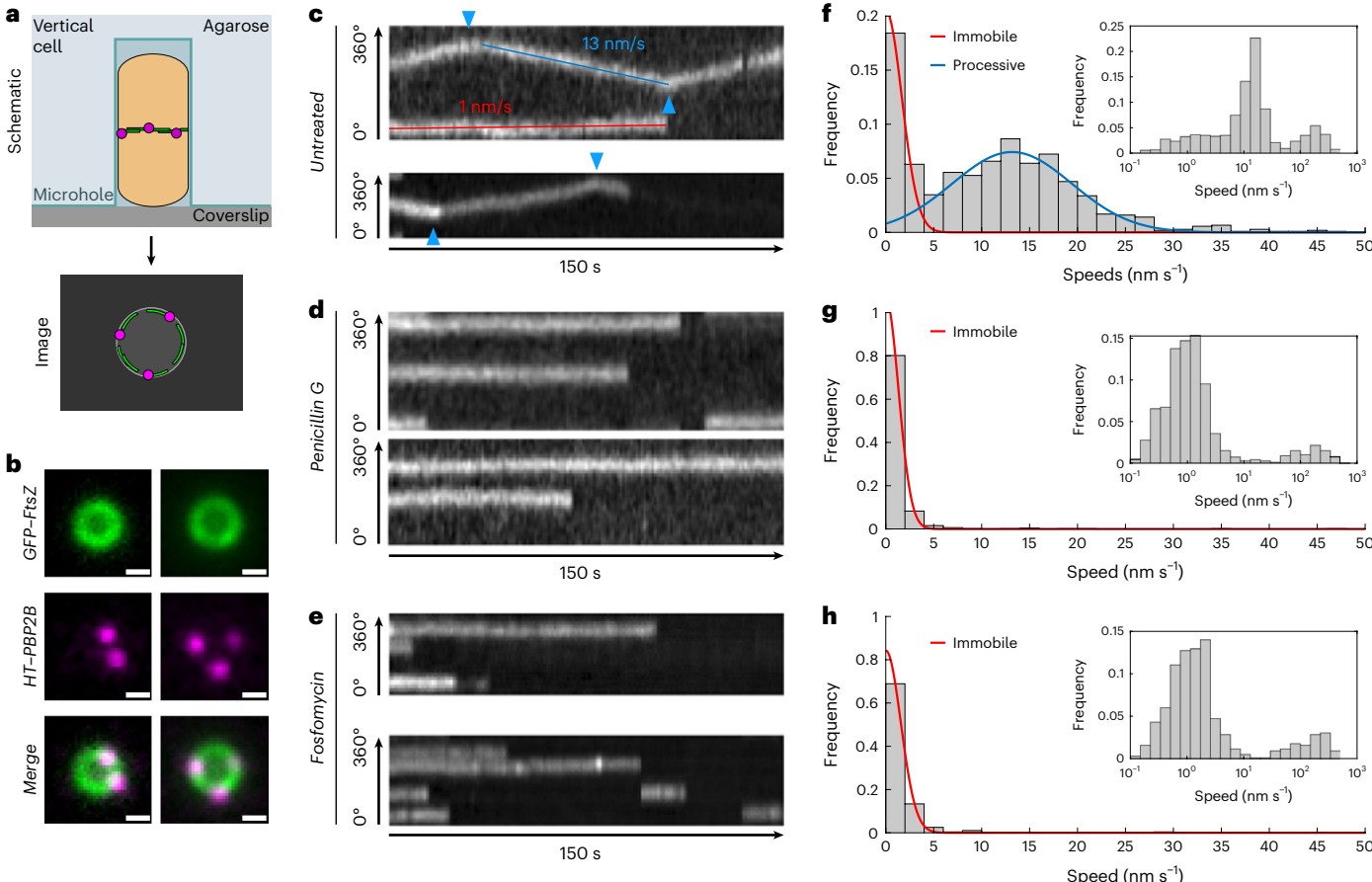

**Fig. 1 | Two-colour smVerCINI imaging shows a single processive population of HT–PBP2B that is associated with sPG synthesis. a**, Schematic of two-colour smVerCINI experimental setup. A *B. subtilis* cell (orange) is oriented vertically in a microhole made of agarose (light blue), adjacent to a microscope coverslip (grey). By orienting the cell vertically, the division proteins FtsZ (green) and PBP2B (magenta) can be imaged in a single microscope imaging plane. **b**, Example images of two vertically trapped cells. GFP–FtsZ (green) is expressed at a low level (0.075% xylose induction) for use as a septal marker, while HT–PBP2B (100 μM IPTG induction) is labelled substoichiometrically with JFX554 HT ligand[17] (250 pM) so that single molecules can be easily observed. Scale bars, 500 nm. **c**, Example radial kymographs of HT–PBP2B motion from smVerCINI videos in untreated cells. Diagonal lines result from processive motion (example shown with blue line), while horizontal lines result from lack of motion (example shown with red line). Blue arrows indicate where HT–PBP2B molecules changed direction. **d,e**, Example radial kymographs of HT–PBP2B motion from smVerCINI videos in cells treated with 20 μg ml⁻¹ penicillin G (**d**) or 500 μg ml⁻¹ fosfomycin (**e**). **f**, Histogram of HT–PBP2B speeds in untreated cells. Red and blue lines show fits to the data (Methods). Inset: histogram of speeds plotted on logarithmic *x* axis, showing three populations. **g,h**, Histograms of speeds in cells treated with penicillin G (**g**) or fosfomycin (**h**). Red lines show fits to the data (Methods). Insets: histograms of speeds plotted on logarithmic *x* axis, showing two populations. All examples in **c**–**h** are from cells grown in rich media at 30 °C. Sample sizes are listed in Supplementary Table 6.

principally exist as part of large multi-protein complexes (that is, the divisome core complex). Alternatively, diffusive HT–PBP2B molecules may experience substantial molecular friction through transient interactions with the cell wall.

**The divisome synthesis complex and FtsZ move asynchronously**

The speed distributions for HT–PBP2B we measured (Fig. 1) are not consistent with either the coupled-complex[2] or two-track[11] models. Both models predict that there exists at least one processive population of synthesis complexes that move at the same speed as treadmilling FtsZ filaments[2,11], but the speed of the processive HT–PBP2B population (13.4 nm s⁻¹ mean; 95% CI 12.7–14.1) (Fig. 1f) does not match what we previously measured for FtsZ treadmilling (44.1 nm s⁻¹ median; 95% CI 26.0–53.9)[9] under identical growth conditions (rich media, 30 °C). This result does not depend on protein expression levels, as inducing HT–PBP2B with a tenfold higher concentration of IPTG resulted in similar speeds for processive HT–PBP2B molecules (Supplementary Fig. 8). Furthermore, in contrast to predictions from the two-track

model[11], we do not observe the emergence of a motile population of HT–PBP2B associated with FtsZ treadmilling after treating cells with penicillin G or fosfomycin (Fig. 1g,h).

Next, we investigated whether the asynchronous motions of synthesis complexes and FtsZ treadmilling are a general feature in *B. subtilis* by repeating our measurements under different growth conditions (Fig. 2a). Our results show that HT–PBP2B speeds depended on both media composition and temperature: processive molecules moved faster in rich media or 37 °C than in minimal media or 30 °C (blue lines in Fig. 2a). Under most growth conditions tested, the speeds of processive HT–PBP2B molecules do not match the speeds of FtsZ treadmilling measured under identical conditions[9] (green dotted lines in Fig. 2a). However, under the fastest growth condition (rich media, 37 °C) HT–PBP2B processive speeds overlap substantially with FtsZ treadmilling speeds. Notably, this was the growth condition under which single-molecule tracking of HT–PBP2B was performed previously[2]. The similarity of speeds between FtsZ treadmilling and HT–PBP2B processive motion in those conditions led us to initially propose that the two systems moved together as a coupled complex. Our results

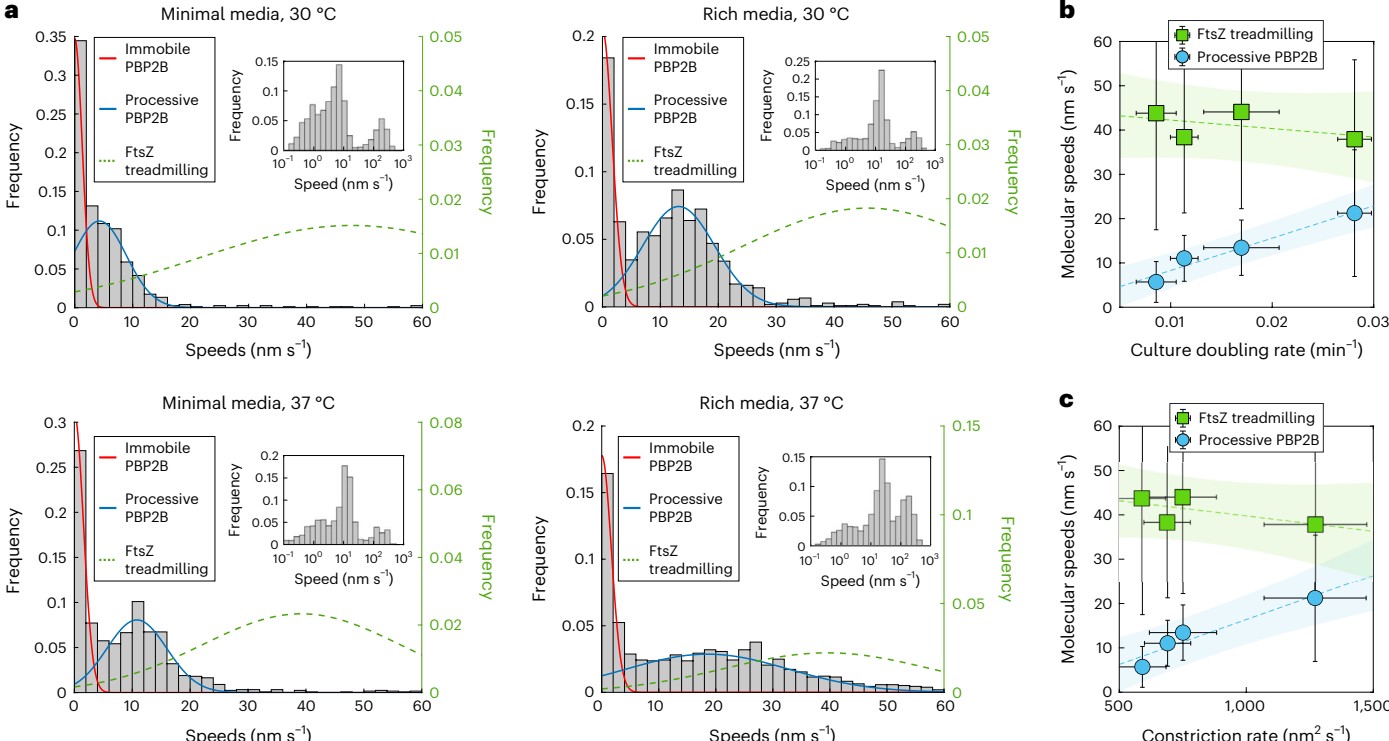

**Fig. 2 | PBP2B and FtsZ move asynchronously. a**, Histograms of HT–PBP2B speeds across growth conditions. Red and blue lines show fits to the data (Methods). Green dotted lines show Gaussian distributions based on the mean and variance of FtsZ treadmilling speeds measured under the same growth conditions (data from Whitley et al.[9]). Insets: histograms of speeds plotted on logarithmic *x* axes, showing three populations. **b**, Correlation between culture doubling rates with speeds of FtsZ treadmilling and processive HT–PBP2B. Dotted lines show linear fits to the data. Shaded regions show mean fitted parameters ± 95% confidence bands. Data for HT–PBP2B speeds and culture

doubling times are presented as mean ± standard deviation (s.d.), while data for FtsZ treadmilling speeds are presented as median ± s.d. **c**, Correlation between FtsZ treadmilling and processive HT–PBP2B speeds with septal constriction rate. Septal constriction rates under these conditions are from Whitley et al.[9]. Dotted line shows a linear fit to the data. Shaded region show mean fitted parameters ± 95% confidence bands. Data for HT–PBP2B speeds and constriction rates are presented as mean ± s.d., while data for FtsZ treadmilling speeds are presented as median ± s.d. Sample sizes and numbers of experimental replicates are listed in Supplementary Table 6.

instead indicate that this overlap in speeds is a coincidence arising only under the fastest growth condition (Fig. 2a).

The variance of the processive HT–PBP2B population under the fastest growth condition is substantially higher than that of other conditions (Fig. 2a). The reason for this increase is unclear, although it was reproducible: each of the four biological replicates we performed under this condition had a similarly large variance. This increase is larger than the dependence predicted by a Poisson process, which is commonly observed in single molecule dynamics. Furthermore, it does not seem to result from perturbed FtsZ treadmilling in the particular strain used here, as the speed distribution for FtsZ treadmilling under these conditions was comparable to those of previous measurements (Supplementary Fig. 4). It is possible that the increased variance in HT–PBP2B speeds under this condition reflects increased variation in the local production or local availability of the lipid II substrate used by the divisome synthesis complex. However, this is currently difficult to investigate experimentally.

We found that processive HT–PBP2B speeds are correlated with cell growth rate (doubling rate in liquid culture), while FtsZ treadmilling is independent of growth rate (Fig. 2b). We speculate that faster growth conditions results in more rapid production of cell wall substrate (lipid II), leading to a higher rate of sPG incorporation reactions and hence higher synthesis complex speeds. We further found that processive HT–PBP2B speeds are correlated with the rate of septal constriction, which we measured previously under identical conditions[9] (Fig. 2c). It is possible that higher synthesis complex speeds lead to more sPG

added to the ingrowing septum per unit time, thereby resulting in a higher rate of septal constriction.

It is surprising that FtsZ treadmilling speed is independent of temperature, as it depends on an enzymatic reaction. Using the Eyring equation with activation enthalpies measured previously for *E. coli* FtsZ in vitro[20], we predict that an increase of 30 °C to 37 °C should result in an approximately twofold increase in treadmilling speed (Methods), which we have not observed. To test how far this temperature independence extrapolates, we measured FtsZ treadmilling speed at 21 °C using a previously characterized strain (bWM4; Supplementary Table 1) expressing mNeonGreen–FtsZ from an IPTG-inducible promoter (Supplementary Fig. 9), grown in rich media. Treadmilling speed was 32% slower than it was at 30 °C, although this is a substantially smaller decrease than the 68% predicted from the Eyring equation. This suggests that bacterial cells may actively regulate the polymerization dynamics of FtsZ filaments, as the lack of temperature dependence of treadmilling speed is not explained by chemical physics alone.

### Processive motion partially depends on FtsZ treadmilling

Our finding that FtsZ and HT–PBP2B move asynchronously suggests that synthesis complexes are independent of FtsZ treadmilling in *B. subtilis*. To test this directly, we measured the dynamics of HT–PBP2B in the absence of FtsZ treadmilling. We treated cells with 10 μM PC190723, an antibiotic that specifically binds to FtsZ (ref. 21) and arrests treadmilling[2] across all stages of division within seconds of treatment[9]. HT–PBP2B still showed processive motion (Fig. 3a,c

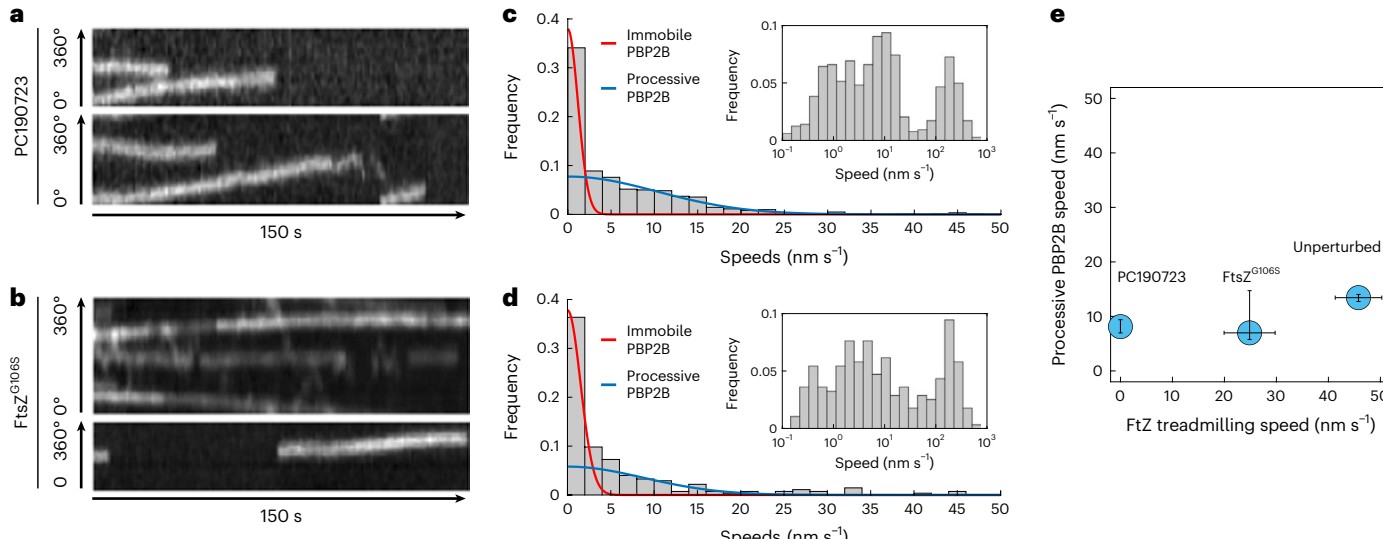

**Fig. 3 | PBP2B motion partially depends on FtsZ treadmilling. a**, Example radial kymographs of HT–PBP2B motion from smVerCINI videos in cells treated with 10 µM PC190723. **b**, Example radial kymographs in cells expressing the treadmilling-impaired mutant FtsZ[G106S]. **c**, Histogram of speeds in cells treated with PC190723. **d**, Histogram of speeds in FtsZ[G106S] cells. Red and blue lines to both histograms show fits to the data (Methods). Insets: histograms of speeds plotted on logarithmic *x* axes, showing three populations. **e**, Relation between FtsZ treadmilling speeds and processive HT–PBP2B speeds. The value for FtsZ[G106S] treadmilling speed was measured in this study under our experimental conditions (Supplementary Fig. 10). Data are presented as mean ± 95% CI. Sample sizes and numbers of experimental replicates are listed in Supplementary Table 6.

and Supplementary Videos 7 and 8), but the speeds were slower than in untreated cells (8.1 nm s⁻¹ mean; 95% CI 7.0–9.4 treated versus 13.4 nm s⁻¹ mean; 95% CI 12.7–14.0 untreated). This is a stark difference from the penicillin G- or fosfomycin-treated cells, where the processive population was abolished (Fig. 1g,h). We confirmed that sPG synthesis continues in the absence of FtsZ treadmilling by treating cells with PC190723 for approximately one round of cell division while simultaneously labelling them with fluorescent D-amino acids, then imaging with structured illumination microscopy (Methods and Extended Data Fig. 5).

We wondered if this drop in speeds was specific to treatment with PC190723, which is a strong perturbation due to its sudden and total arrest of FtsZ treadmilling. We therefore tried a similar genetic perturbation by using an FtsZ mutant (FtsZ[G106S]) that is competent for division but has reduced treadmilling speed[2] (Supplementary Fig. 10) and produces a long-cell phenotype (Methods, Supplementary Table 1 and Supplementary Fig. 11). The speed distribution for HT–PBP2B in this mutant strain was similar to that observed with PC190723-treated wild-type cells (Fig. 3b,d and Supplementary Videos 9 and 10). This suggests that any substantial disruption to FtsZ treadmilling has similar effects on synthesis complex motion (Fig. 3e), in contrast to the linear relation between their speeds proposed previously[2].

We recently reported evidence from computational studies that perturbations to FtsZ treadmilling disrupt the nematic order of the FtsZ filament network[22]. We therefore considered that the reduction in HT–PBP2B speeds could result from motion along a disordered, jagged path due to transient interactions between the divisome synthesis complex and randomly oriented FtsAZ filaments. To test this, we imaged the processive motion of HT–PBP2B molecules in horizontally oriented cells and measured the displacements from the septal axis (Methods, Supplementary Videos 14–16 and Extended Data Fig. 6). With either PC190723 treatment or expression of FtsZ[G106S], the median off-axis displacements were within 2 nm of that measured for the unperturbed case, which is unlikely to be biologically meaningful (Extended Data Fig. 6e). This suggests that the reduction in HT–PBP2B speeds observed upon perturbations to FtsZ treadmilling does not result from HT–PBP2B off-axis motion.

## The divisome synthesis complex is multimeric

During this study, we observed many cases where the fluorescence intensities of HT–PBP2B spots showed discrete drops to half their value (Fig. 4a,b and Supplementary Videos 2 and 11), indicating the presence of two copies of the fluorescently labelled protein. Under our standard conditions (rich media, 30 °C, 100 µM IPTG and 250 pM JFX554 HT ligand), we observed these intensity drops in 11% (*N* = 56) of full tracks. We also observed multiple occasions where such intensity drops occurred during motion and even after direction changes (Fig. 4a,b and Extended Data Fig. 7), strongly suggesting that multiple monomers of HT–PBP2B are moving together as part of a larger complex. Due to the substoichiometric nature of the labelling method, we cannot precisely quantify the number of HT–PBP2B molecules in a given complex, although we have observed rare cases with even three or four such drops in intensity (Extended Data Fig. 8).

Surprisingly, we also observed cases where the fluorescence intensity signal shows discrete jumps to twice their value (4% (*N* = 22) of full tracks under our standard conditions; Fig. 4c,d, Extended Data Fig. 9 and Supplementary Videos 12 and 13). We observed these discrete jumps in both immobile and processive tracks. This suggests that the oligomeric state of synthesis complexes is dynamic, where new PBP2B molecules can bind to both active and inactive complexes. As only 3 out of 22 observed intensity jumps (14%) and 9 out of 56 (16%) of observed intensity drops roughly corresponded to a change in HT–PBP2B speed, it appears that these events do not necessarily affect divisome synthesis complex activity, although it remains possible that there is a higher probability of speed change during an intensity drop/jump event than without.

We wondered whether this behaviour was unique to PBP2B, or if it was a more general feature of divisome synthesis complex proteins. We repeated these measurements with a strain expressing a HT fusion of the transglycosylase FtsW (HT–FtsW; Supplementary Table 1) that together with PBP2B forms the core of the synthesis complex. HT–FtsW displayed similar fluorescence drops and jumps as HT–PBP2B (Extended Data Fig. 10), suggesting that the divisome synthesis complex is multimeric. The effect of stoichiometry on divisome activity and dynamics will be followed up in future work.

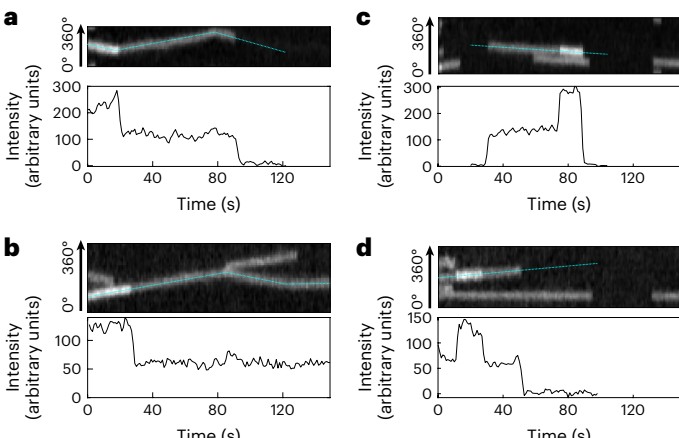

**Fig. 4 | PBP2B is part of dynamic multimeric complexes. a,b,** Examples of discrete drops in fluorescence intensity mid-track. Top: example kymographs of HT–PBP2B motion from smVerCINI videos. Bottom: intensity traces for the tracks designated by dotted cyan lines overlaid on kymographs. Both traces are from cells grown in rich media at 30 °C. **c,d,** Examples of discrete jumps in fluorescence intensity mid-track. Top: example kymographs. Bottom: intensity traces for the tracks designated by dotted cyan lines overlaid on kymographs. Both traces are from cells grown in minimal media at 30 °C.

## Discussion

Our results show that a multimeric divisome synthesis complex in *B. subtilis* follows a single track dependent on sPG synthesis and asynchronous with FtsZ treadmilling. This sharply contrasts with two prominent models for bacterial cell division[2,11], which predict the existence of a processive population of synthesis complexes associated with FtsZ treadmilling. Our results instead support a model of septal PG synthesis where the Z-ring recruits the synthesis complex to the septal leading edge but does not directly regulate its motion and synthesis activity (Fig. 5).

Our finding that the motions of synthesis complexes and FtsZ are asynchronous is consistent with previous measurements in *S. pneumoniae*[5] and work performed in parallel to this study in *S. aureus*[23]. Our discovery that synthesis complexes retain processive motion in the absence of FtsZ treadmilling also agrees with previous findings in both *S. aureus* and *B. subtilis* that sPG synthesis continues in the absence of FtsZ treadmilling if constriction has already initiated[4,9]. This suggests that these species—all members of the phylum Bacillota (Firmicutes)—share a similar mechanism for division. It is possible that the two-track model proposed for *E. coli* and *C. crescentus*—both members of the phylum Pseudomonadota (Proteobacteria)—represents a special case arising from requirements unique to this bacterial clade, such as a thinner PG layer or sparser network of FtsZ filaments.

Previously, our collaborators observed that arrest of FtsZ treadmilling by the antibiotic PC190723 abolished processive HT–PBP2B motion[2], but here we find that the effect of arresting FtsZ treadmilling is to slow HT–PBP2B processive motion rather than stop it. One difference between these studies is that cells were imaged here with VerCINI while the previous study imaged horizontally oriented cells with total internal reflection fluorescence (TIRF) illumination. However, as part of this study we also imaged horizontally oriented cells with TIRF and observed processive motion at division septa post-PC190723 treatment (Supplementary Video 15). It is possible that processive motion was not observed in the previous study due to small sample size.

We observed that perturbations to FtsZ treadmilling lead to slower speeds for processive synthesis complexes. The reduction in speeds upon total arrest of FtsZ treadmilling (~40%) correlates with the reduction in septal constriction rate (by 14–33%, depending on growth

conditions) that we previously observed under similar conditions[9]. This is corroborated by our finding that synthesis complex speeds are correlated with septal constriction rate (Fig. 2c). However, work performed in parallel to this study in *S. aureus* found that synthesis complex speeds and septal constriction rate are not affected by FtsZ treadmilling[23]. In contrast to both of these studies, previous work in *S. pneumoniae* found a minor (~25%) reduction in synthesis complex speeds when FtsZ treadmilling was severely perturbed, and no effect from a smaller perturbation[5]. This suggests that, although these three species may share a similar mechanism for division, there may remain differences, which could arise from changes in the interaction strength of FtsZ subunits with the rest of the divisome.

The underlying mechanism for the reduction in synthase complex speeds upon FtsZ treadmilling perturbation remains unclear. It is possible that the disruption to FtsZ treadmilling leads to an alteration of transient interactions between FtsAZ filaments and the divisome synthesis complex, leading to increased molecular friction. However, we consider this unlikely, as both a reduction and total arrest of FtsZ treadmilling produced similar effects on HT–PBP2B speeds (Fig. 3e). The reduction in HT–PBP2B speeds may instead be an indirect effect of disrupting FtsZ treadmilling. We have shown that the speeds of the divisome synthesis complex depends on cell metabolism (Fig. 2b). It is plausible that severe perturbations to an essential and abundant protein such as FtsZ could affect metabolism (for example, through a stress response) and hence indirectly reduce synthase speeds.

Our results, along with those in multiple other organisms[5,23], strongly support a model where the processive motion of sPG synthases in Bacillota is driven exclusively by sPG synthesis. Our observation that treatment with antibiotics targeting either synthase activity

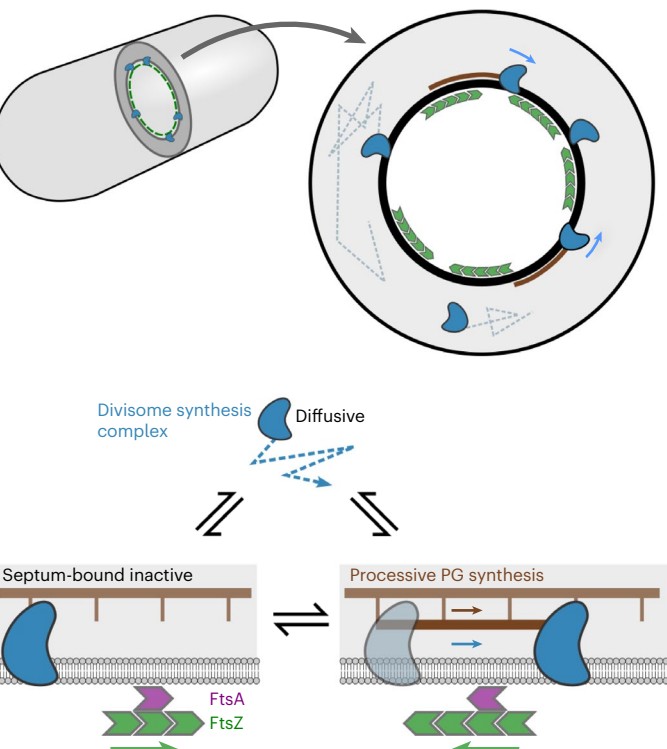

**Fig. 5 | A one-track model for spatiotemporal regulation of *B. subtilis* sPG synthesis activity.** Inactive divisome synthesis complexes (blue) diffuse around the cell membrane and are recruited to the septal leading edge, marked by a dense FtsZ-ring (green), via transient interactions with the FtsZ anchors FtsA (purple) and SepF (not shown for clarity). Once recruited to the septum, synthesis complexes move asynchronously from FtsZ, switching between an inactive static state, potentially bound to the cell wall, and an active state performing processive septal PG synthesis.

or lipid II precursor synthesis prevents processive motion (Fig. 1c–h) suggests that the sPG insertion reaction itself may provide the required energy. A similar mechanism was previously proposed for elongasome synthases[24–26]. Further work is required to understand the molecular mechanism by which PG synthesis activity by SEDS-bPBP (shape, elongation, division and sporulation – class B penicillin binding protein) complexes leads to processive motion.

We found evidence that the divisome synthesis complex—the motile multi-protein complexes of divisome proteins that move around the septum synthesizing PG—is multimeric, in support of the stoichiometric divisome hypothesis[27]. In *P. aeruginosa*, the structure of the so-called divisome core complex of proteins, which is probably the minimum holoenzyme unit required to synthesize and attach a single strand of septal PG, was reported[6]. We found that AlphaFold predicts a *B. subtilis* divisome core complex very similar to that of *P. aeruginosa* (Extended Data Fig. 1), consisting of PBP2B (*P. aeruginosa* FtsI/ PBP3 homologue) in a complex with FtsW, DivIC (FtsB homologue), FtsL and DivIB (FtsQ homologue). As the stoichiometry of PBP2B and FtsW with each of the other components in the divisome core complex is 1:1, our data suggest that individual divisome synthesis complexes often contain multiple divisome core complexes. The multimeric nature of the divisome synthesis complex may also explain the abrupt changes in direction we observe in single-molecule tracks (Fig. 1c). Such bidirectional motion could result from multiple synthesis proteins in a complex pulling in opposite directions, as is well known for motor proteins moving along the eukaryotic cytoskeleton[28] and more recently the elongasome of *B. subtilis*[16]. Inclusion in a large multimeric complex may also explain the very low effective diffusion coefficient we measured for HT–PBP2B in the septal ring (Extended Data Fig. 4e). These possibilities will be investigated in future studies.

## Methods

### Divisome complex modelling
The divisome complex was modelled by AlphaFold2, using ColabFold (v1.3.0) and AlphaFold-Multimer (v2). The sequences were downloaded from the UniProtKB database and included five divisome protein sequences (Q07868 (PBP2B), Q07867 (FtsL), O07639 (FtsW), P16655 (DivIB) and P37471 (DivIC)).

### Bacterial strains and growth conditions
Strains used in this study are listed in Supplementary Table 1. Strains were streaked from −80 °C glycerol stocks onto nutrient agar plates containing the relevant antibiotics and 1 mM IPTG and grown overnight at 37 °C. Single colonies were transferred to liquid starter cultures in either rich medium (PHMM)[2] or time-lapse medium[29] (TLM) with required inducers, shaking at 200 rpm overnight at either 22 °C or 30 °C. The next day, PHMM starter cultures were diluted to $OD_{600}$ (optical density at 600 nm) of 0.05 in fresh PHMM, while TLM starter cultures were diluted to $OD_{600}$ of 0.1 in chemically defined medium[29]. These liquid cultures were grown at 30 °C or 37 °C with required inducers until they reached $0.3 < OD_{600} < 0.6$. When necessary, antibiotics were used at the following concentrations: spectinomycin 60 µg ml⁻¹, erythromycin 1 µg ml⁻¹, lincomycin 25 µg ml⁻¹ and 6 µg ml⁻¹ tetracycline.

### Strain construction
Strains SH147 and SH203 harboured a deletion of the *hag* gene, as this has been shown to reduce the chaining phenotype in *B. subtilis* cells and thereby increase loading into micro-holes[9,15].

SH142 (PY79 Δ*hag, amyE*::*spc*-P_xyl-*gfp-ftsZ*) was constructed by transforming SH211 with genomic DNA extracted from strain 2020 using standard protocols[30].

SH147 (PY79 Δ*hag, pbpB*::*erm*-P_hyperspank-*HT-15aa-pbpB, amyE*::*spc*-P_xyl-*gfp-ftsZ*) was constructed by transforming SH142 with a PCR product obtained from bGS31 genomic DNA. The primer pair ftsL Fw (5′-ATGAGCAATTTAGCTTACCAACC-3′) and spoVD

Rev (5′-TCAATCGGCTGCCTCCTTTTC-3′) were used to amplify the region around the *pbpB* gene using bGS31 genomic DNA as a template. The primer pair ftsL Fwd/pbp2B Rev (5′-TTAATCAGGATTTTTAAACTTAACCTTGATTACGG-3′) was used to confirm insertion of the *HT* gene in the transformant. Insertion was also confirmed by Sanger sequencing. Primers were from Integrated DNA Technologies.

SH203 (PY79 Δ*hag, pbpB*::*erm*-P_hyperspank-*HT-15aa-pbpB, ftsZΩftsZ(G106S) (tet), amyE*::*spc*-P_xyl-*gfp-ftsZ*) was constructed by transforming SH147 with genomic DNA extracted from strain Z-G106S (gifted by Ethan Garner). The point mutation G106S was confirmed by Sanger sequencing.

All published strains are available on request to the authors.

### Bacterial strain characterization
Strains were characterized by growth in liquid culture (Supplementary Figs. 3 and 11) and cell morphology analysis (Supplementary Figs. 1 and 11).

### Growth curves
*B. subtilis* PY79 and variant strains were grown in liquid starter cultures overnight in lysogeny broth (LB) at 30 °C with required inducers (100 µM IPTG for SH147 and SH203). To measure culture growth across (IPTG) (Supplementary Fig. 3), SH147 overnight cultures were washed twice in LB to remove inducer. Cultures were then diluted to $OD_{600}$ of 0.05 in LB with variable inducer concentrations, and 200 µl of each dilution was added to a 96-well microtitre plate. Growth was monitored for 15 h using a FLUOStar OPTIMA plate reader (BMG Labtech). Growth curves for each condition were performed in triplicate.

### Western blotting
Overnight cultures of specified strains were grown overnight in PHMM at 22 °C. The following morning, cultures were diluted to an $OD_{600}$ of ~0.05 and grown at 37 °C until an $OD_{600}$ of ~0.4. Cells were collected by centrifugation and lysed by incubation for 20 min in BugBuster protein extraction reagent supplemented with Benzonase nuclease (Millipore) and an EDTA-free protease inhibitor cocktail (Roche). Protein extract was heated for 10 min at 65 °C in NuPAGE LDS Sample Buffer, then 5 µg total protein in this buffer was separated by sodium dodecyl sulfate polyacrylamide gel electrophoresis on a NuPAGE 3–8% Tris-acetate Midi gel (Invitrogen). Protein was transferred to a 0.45 µm PVDF membrane (Cytiva), and PBP2B and Spo0J were detected using PBP2B polyclonal (1:5,000 dilution) and Spo0J polyclonal (1:2,500 dilution) antibodies, respectively, followed by a horseradish peroxidase-conjugated anti-rabbit IgG antibody (Sigma; 1:10,000 dilution). Samples were developed using Clarity Western ECL Substrate (Bio-Rad) and imaged using an ImageQuant LAS 4000 mini Biomolecular Imager (GE Healthcare).

### Cell morphology analysis
*B. subtilis* Δ*hag* (strain SH211) and variant strains were grown in liquid starter cultures overnight in PHMM at 22 °C or 30 °C with required inducers (100 µM IPTG for SH147 and SH203). To measure cell morphology across IPTG concentrations (Supplementary Fig. 1c), SH147 overnight cultures were washed twice in PHMM to remove inducer. Cultures were then diluted to an $OD_{600}$ of 0.05 in PHMM with variable IPTG concentrations and 0.075% xylose. To measure cell morphology across xylose concentrations (Supplementary Fig. 1d), overnight SH147 cultures were diluted to an $OD_{600}$ of 0.05 in PHMM with 100 µM IPTG and variable xylose concentrations. Once the cultures had reached $0.3 < OD_{600} < 0.6$, Nile Red was added to cells and incubated at growth temperatures for 5 min. Then 0.5 µl of cell culture was spotted on 1.2–2% agarose pads of PHMM with the required inducers, prepared as described previously[29]. Cells were imaged using a 561 nm laser or 550 nm light-emitting diode, and cell lengths were manually determined using ImageJ and MATLAB.

## smVerCINI

smVerCINI was set up as described previously for VerCINI[9,15]. Briefly, agarose microholes were formed by pouring molten 6% agarose onto a nanofabricated silicon array consisting of micropillars with widths 1.0–1.3 μm and heights 6.8 μm. Patterned agarose was transferred into a Geneframe (Thermo Scientific) mounted on a glass slide, and excess agarose was cut away to ensure sufficient oxygen. Labelled cells at $0.3 < OD_{600} < 0.6$ were concentrated 100× by centrifugation and added onto the agarose pad. Cells were then loaded into the microholes by centrifuging the mounted agarose pad with concentrated cell culture in an Eppendorf 5810 centrifuge with MTP/Flex buckets. Unloaded cells were rinsed off with excess media. In experiments where cells were treated with antibiotic (Figs. 1d,e and 3a), 5 μl of media laced with antibiotic was added to the top of loaded cells and allowed to absorb for ~1 min before sealing with a coverslip and imaging.

Imaging was done by first recording a single frame of GFP–FtsZ using the 488 nm laser to identify division rings. Immediately following this, a time lapse of HT–PBP2B dynamics was recorded using the 561 nm laser. Following fluorescence imaging, a short bright-field video was recorded to identify any cells that were improperly trapped in micro-holes. Microscopy acquisition parameters are listed in Supplementary Table 5.

## HT labelling with JFX554 dye

Single-molecule labelling of HT–PBP2B was done by incubating strain SH147 or SH203 with either 100 pM (minimal media) or 250 pM (rich media) JFX554 HT ligand for 15 min unless otherwise noted. Cells were washed once with fresh media before imaging. JFX554 HT ligand was a gift from Luke Lavis (Janelia Farm)[17].

## Microscopy

Power densities, exposure times and other key parameters are listed for each microscopy experiment in Supplementary Table 5.

**Nikon Eclipse Ti2.** Cells were illuminated with 488 nm and 561 nm laser illumination. A 100× TIRF objective (Nikon CFI Apochromat TIRF 100XC Oil) was used for imaging, and a Kinetix sCMOS camera (Teledyne Photometrics) was used with effective pixel size of 65 nm per pixel. Cells were illuminated using highly inclined and laminated optical sheet (HiLO)[31] or TIRF to minimize background using an objective TIRF module. Acquisition was performed using NS-Elements (v5.42.02).

**Bespoke microscope.** Cells were illuminated with a 488 nm laser (Obis) and a 561 nm laser (Obis). A 100× TIRF objective (Nikon CFI Apochromat TIRF 100XC Oil) was used for all experiments. A 200 mm tube lens (Thorlabs TTL200) and Prime BSI sCMOS camera (Teledyne Photometrics) were used for imaging, giving an effective pixel size of 65 nm per pixel. Imaging was done with a custom-built ring-TIRF module operated in ring-HiLO using a pair of galvanometer mirrors (Thorlabs) spinning at 200 Hz to provide uniform, high SNR illumination[32]. Acquisition was performed using Micro-Manager (v2.0 gamma).

**Nikon Eclipse Ti.** Cells were illuminated with a 550 nm light-emitting diode (CoolLED). A 100× TIRF objective (Nikon Plan Apo 100×/1.40 numerical aperture Oil Ph3) was used with a Prime BSI camera (Teledyne Photometrics). Acquisition was performed using NS-Elements (v5.42.02).

**Zeiss Elyra 7 Lattice SIM2.** Cells were illuminated with 488 nm and 561 nm lasers. A 63× objective (Plan Apo 63×/1.40 Oil) was used for structured illumination microscopy (SIM) experiments. A 1.6× Optovar and two PCO.edge 4.2 sCMOS cameras (PCO Imaging) were used for imaging, giving an effective pixel size of 62 nm per pixel. Acquisition was performed using Zen Black 2.3 (v16.0.14.316).

## TIRF microscopy of horizontal cells

Coverslips were first cleaned by treating with air plasma for 5 min. Slides were prepared as described previously[29]. Flat 2% agarose pads of PHMM containing inducers were prepared inside Geneframes (Thermo Scientific) and cut down to strips of ~5 mm width to ensure sufficient oxygen supply to cells. Cell cultures were grown to $OD_{600}$ between 0.4 and 0.7, when 0.5 μl of cell culture was spotted on the pad. Cells were allowed to adsorb to the pad for ~1 min. In the case where the HT–PBP2B GFP–FtsZ Δhag strain (SH147) was treated with PC190723, 1 μl of a solution of PHMM + 100 μM IPTG + 0.075% xylose + 10 μM PC190723 was then spotted on top of the cells and allowed to absorb into the pad for ~1 min. A plasma-treated coverslip was then placed on top. Cells were allowed to equilibrate within the microscope body for ~2 min before being imaged. Cells were then imaged using TIRF microscopy to observe either FtsZ treadmilling dynamics or HT–PBP2B dynamics. If the concentration of labelled FtsZ was too high to measure treadmilling speeds, the illumination mode was changed to HiLO for 1–10 s to photobleach the label down to an acceptable level before data were acquired. Experimental parameters are defined in Supplementary Table 5.

## Structured illumination microscopy of cells labelled with fluorescent D-amino acids

*B. subtilis* PY79 and variant strains were grown in liquid starter cultures overnight in PHMM at 30 °C with required inducers (100 μM IPTG for SH147 and SH203). The overnight cultures were then diluted to an $OD_{600}$ of 0.1 into fresh PHMM and grown at 30 °C until $0.4 < OD_{600} < 0.6$. At this point, cultures were re-diluted to an $OD_{600}$ of 0.1 in pre-warmed PHMM, and 200 μl of diluted culture was transferred to 2 ml tubes with holes in the lid for aeration. The green fluorescent D-amino acid BODIPY-FL 3-amino-D-alanine (BADA)[33] (Tocris Bioscience) was added to a final concentration of 0.5 mM, and tubes were incubated at 30 °C with shaking for 90 min. Samples were washed with 200 μl pre-warmed PHMM. After the second wash, the red fluorescent D-amino acid TAMRA-amino-D-alanine (TADA)[34] (Tocris) was added to a final concentration of 0.5 mM. Where cells were treated with PC190723, this compound was also added to a final concentration of 14 μM. Samples were then re-incubated at 30 °C with shaking for 10 min. Cells were then washed once with pre-warmed PHMM before the addition of 100% ice-cold ethanol. Samples were fixed on ice for 1 h. Fixed cells were collected by centrifugation and washed twice with cold phosphate-buffered saline.

A total of 0.5 μl cells were spotted onto 2% agarose pads in Geneframes (Thermo Scientific) and allowed to adsorb for several minutes before the addition of a coverslip. Cells were imaged by two-dimensional SIM on an Elyra 7 Lattice SIM2 microscope (Zeiss). All images were acquired with an exposure time of 118 ms and a laser power of 2% in each channel. Alignment of the two imaging channels was conducted using Tetraspeck fluorescent beads as fiducial markers (Invitrogen). SIM image processing was performed in Zen Black, in two-dimensional SIM mode, using the standard configuration. Image registration to correct for residual misalignment between the two imaging channels was performed using Zen Black Channel Alignment tool, fitting an Affine transform between the two imaging channels based on image similarity and then applying it to the red image channel. After SIM reconstruction, images had an effective pixel size of 31 nm.

## Image processing and analysis

Videos were denoised using the GPU-accelerated ImageJ plugin PureDenoise-GPU (v0.1.0)[16] or the CPU-based version PureDenoise-CPU (v0.1.0)[35].

Cells with mature or constricting division rings in focus were chosen using the first GFP–FtsZ image acquired in the imaging sequence described above. The short bright-field videos of each field of view acquired after fluorescence imaging were then used to filter out any of these cells that were improperly trapped in the holes.

Previously developed software[9,15] was used to subtract the cytoplasmic background signal and produce radial kymographs. Due to chromatic aberration, the HT–PBP2B signal itself was used for fitting and extracting radial kymographs rather than the single GFP–FtsZ frame. Due to the difficulty of fitting a circle to these sparsely labelled rings, the maximum intensity projections of the HT–PBP2B videos were used for fitting, and each fit was manually inspected to confirm it was adequate.

Measurements of filament speed and processivity were performed manually by kymograph analysis, annotating filaments as lines in ImageJ and then measuring the angle via ImageJ script. For visual display purposes, kymographs represented here have the circle origin (0°/360°) rotated around the division ring so that single-molecule tracks are appropriately shown as continuous.

### Speed distribution analysis

Speed histograms were fitted to a sum of Gaussian distributions. For most histograms, a sum of three Gaussian distributions was used, but in the cases of penicillin G and fosfomycin treatment (Fig. 1g,h) these fits yielded two overlapping distributions at 0 nm s$^{-1}$. In these two cases, then, a sum of two Gaussian distributions was used rather than three. As one population was expected to represent fully immobile molecules with speed of 0 nm s$^{-1}$, this one parameter was fixed in all cases. Processive HT–PBP2B speeds were determined from these fits by calculating the first moment of the Gaussian distribution fitting the processive population. The 95% CIs were obtained by bootstrapping.

### Single-particle tracking and MSD analysis

For analysis of the diffusive HT–PBP2B population, single molecules were detected and tracked using TrackMate[36] with linking distance of 0.5 μm, five frame gaps and 0.5 μm gap-closing distance. Only tracks more than ten frames long were used for further analysis. Bespoke MATLAB code was used to plot tracks in polar coordinates and fit MSDs to the anomalous diffusion model MSD = $4D_{eff}t^{\alpha}$, where $D_{eff}$ is the effective diffusion coefficient and $\alpha$ is the anomalous diffusion exponent.

### Calculation of transition rates

For a state A that can transition to multiple other states B, C, D and so on, where each competing transition is associated with a distinct rate constant $k_B$, $k_C$, $k_D$ and so on:

The rate of leaving state A overall is given by a sum of all competing rates:

$$k_A = k_B + k_C + \cdots$$

The rate of depletion of A is given by

$$\frac{dA}{dt} = -k_A A(t)$$

while the rate of accumulation of B is given by

$$\frac{dB}{dt} = k_B A(t)$$

and similarly for the accumulation of other states.

The fraction of state A over time is then given by the first-order ordinary differential equation (ODE) as

$$A(t) = k_A e^{-k_A t}$$

and the fraction of state B over time is then given by

$$B(t) = k_B e^{-k_A t}$$

and similarly for the fractions of other states.

If we want to know the total fraction that ended up in state B, we integrate over all time:

$$\int_{t=0}^{\infty} B(t')\, dt' = \int_{t=0}^{\infty} k_B e^{-k_A t'} dt'$$

yielding

$$B = \frac{k_B}{k_A} = k_B \langle t_A \rangle$$

where $\langle t_A \rangle$ is the mean lifetime of state A.

This means that we can calculate transition rate $k_B$ (and all others) using the fraction of events ending in state B and the measured lifetime of state A:

$$k_B = \frac{B}{\langle t_A \rangle}$$

### Temperature dependence of FtsZ treadmilling

We can predict the effect of temperature on the rates of polymerization and depolymerization using the Eyring equation:

$$k = \frac{k_B T}{h} e^{-\frac{\Delta H^{\ddagger}}{k_B T} + \frac{\Delta S^{\ddagger}}{k_B}}$$

If we are comparing the rates across two temperatures, we can rearrange this expression to

$$\frac{k_1}{k_2} = \frac{T_1}{T_2} e^{-\frac{\Delta H^{\ddagger}}{k_B}\left(\frac{1}{T_1} - \frac{1}{T_2}\right)}$$

along with estimates for $\Delta H^{\ddagger}$ measured for *E. coli* FtsZ GTPase/depolymerization in vitro[20]. Using the published value of $\Delta H^{\ddagger} = 98.4$ kJ mol$^{-1}$ for depolymerization, we find that an increase from 30 °C to 37 °C should cause an increase of approximately twofold in this rate, and hence probably in treadmilling speed. Similarly, a decrease from 30 °C to 21 °C should cause a drop of ~63% in speed.

### Statistics and reproducibility

Fitting and plotting of data throughout the paper was done with bespoke MATLAB code (see 'Code availability'). All data shown in violin plots were analysed using Data Analysis with Bootstrap Coupled Estimation (DABEST)[37] to show magnitude and robustness of effect rather than simply statistical significance, using previously described bespoke MATLAB analysis code[38]. Sample sizes were calculated by summing total numbers of data points. All sample sizes and number of experimental replicates can be found in Supplementary Table 6.

### Reporting summary

Further information on research design is available in the Nature Portfolio Reporting Summary linked to this article.

## Data availability

The sequences for performing protein structure predictions were downloaded from the UniProtKB database (Q07868 (PBP2B); Q07867 (FtsL); O07639 (FtsW); P16655 (DivIB); P37471 (DivIC)). Source data for all figures presented in the paper and Supplementary Information, as well as representative raw video data, are available at https://doi.org/10.25405/data.ncl.c.7078312 ref. 39.

## Code availability

Custom software is available on the Whitley lab GitHub page or Zenodo at https://github.com/WhitleyLab/Vercini_spt_analysis ref. 40.

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

## Acknowledgements

We acknowledge E. Garner (Harvard) for sharing strains, L. Juan Wu (Newcastle) for sharing strains and antibodies, L. Lavis (Janelia Farm) for providing us with JFX554 HT ligand and Stuart Middlemiss (Newcastle) for advice with cloning. We also thank M. Pinho (ITQB NOVA) and J. Xiao (Johns Hopkins) for helpful discussions. This work was funded by a Newcastle University Research Fellowship (S.H.), a Wellcome Trust and Royal Society Sir Henry Dale Fellowship (206670/Z/17/Z) (S.H.), a BBSRC 19ALERT mid-range equipment initiative grant (BB/T017570/1) (S.H.) and awards from the Wellcome Trust (208361/Z/17/Z) (P.J.S.),

# Article

MRC (MR/S009213/1) (P.J.S.) and BBSRC (BB/P01948X/1, BB/R002517/1 and BB/S003339/1) (P.J.S.). We acknowledge use of a Zeiss Lattice SIM2 within the School of Life Sciences Microscopy Facility, University of Warwick and a BBSRC ALERT grant (BB/W020300/1) (S.H.), which supported purchase of that microscope.

## Author contributions

K.D.W. and S.H. designed the research; K.D.W. and D.M.R. performed the experiments; K.D.W., J.G. and E.K. constructed and characterized bacterial strains; K.D.W., D.M.R. and J.G. analysed the data. P.J.S. performed protein structure predictions; K.D.W. and S.H. wrote the manuscript with input from all authors.

## Competing interests

The authors declare no competing interests.

## Additional information

**Extended data** is available for this paper at https://doi.org/10.1038/s41564-024-01650-9.

**Correspondence and requests for materials** should be addressed to Kevin D. Whitley or Séamus Holden.

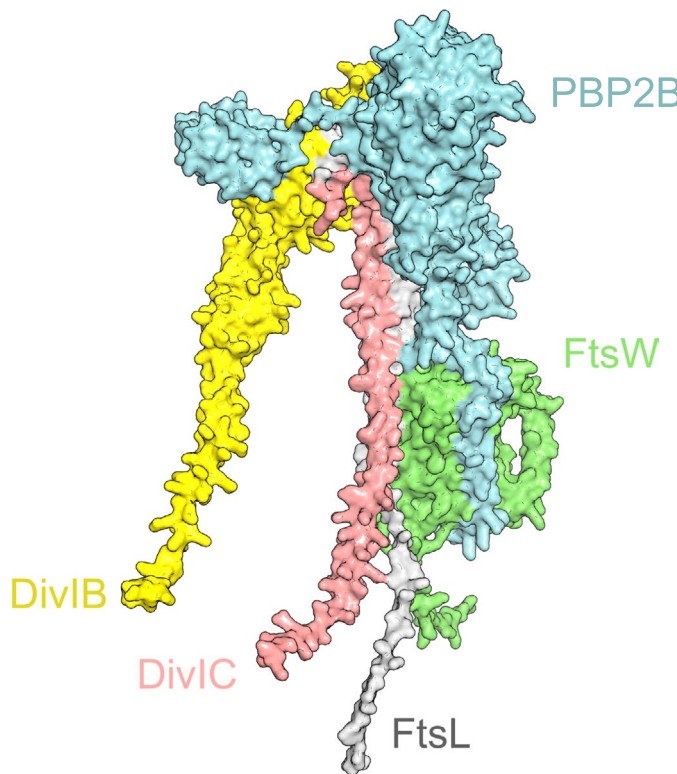

**Extended Data Fig. 1 | Model of the _B. subtilis_ divisome synthesis complex.** Model of the pentameric _B. subtilis_ divisome complex consisting of five proteins (PBP2B, cyan; FtsW, green; FtsL, grey; DivIC, pink; DivIB, yellow). Modelling details are described in Methods.

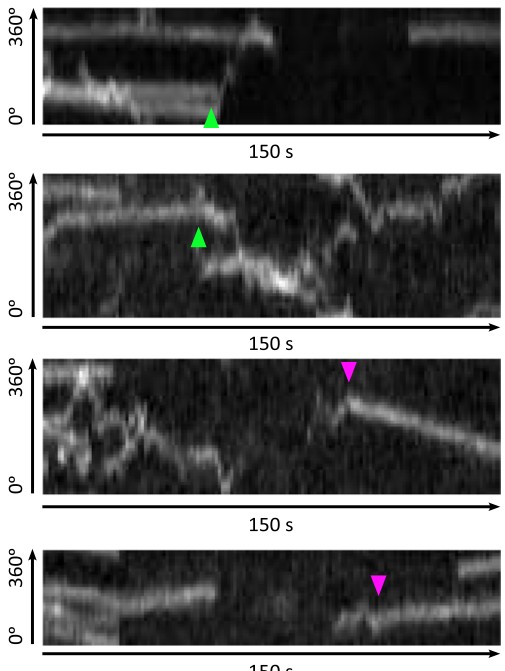

**Extended Data Fig. 2 | Transitions to and from the diffusive state.** Four example radial kymographs showing HT-PBP2B molecules transitioning from either an immobile or processive state to a diffusive state (green arrowheads), or from a diffusive state to a processive state (magenta arrowheads).

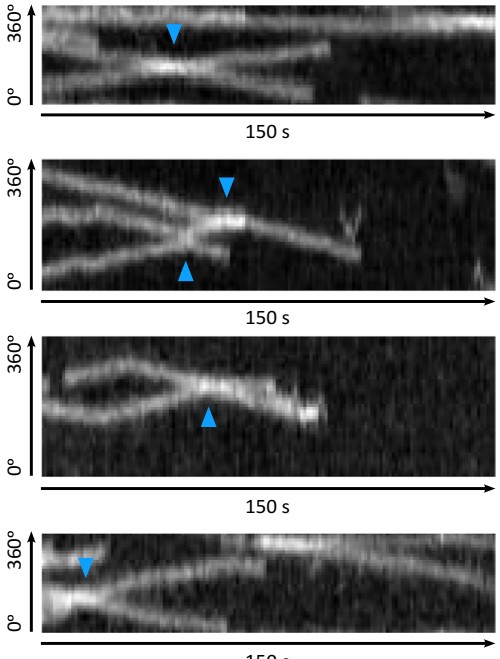

**Extended Data Fig. 3 | Track crossing events.** Four examples of radial kymographs of HT-PBP2B showing processive tracks crossing over one another (crossing events shown with blue arrowheads).

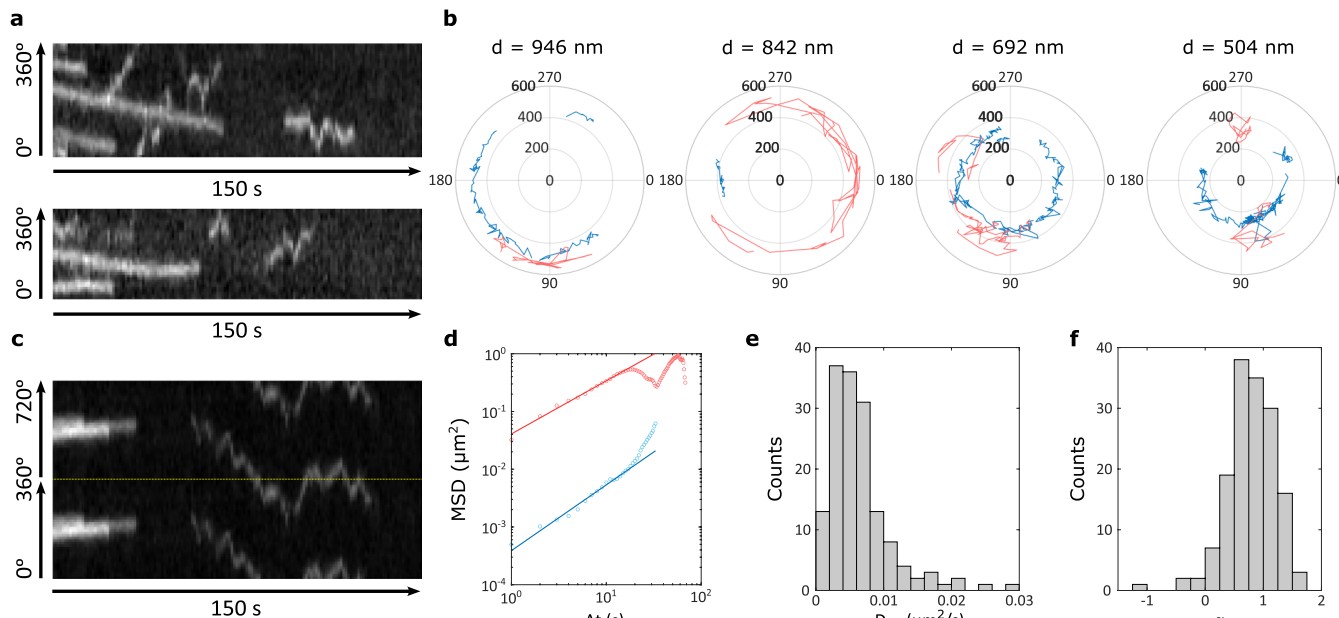

**Extended Data Fig. 4 | Diffusive motion of HT-PBP2B. (a)** Two example radial kymographs of HT-PBP2B (rich media, 30 °C) showing both processive and diffusive tracks. Diffusive tracks appear as short back-and-forth lines. **(b)** Examples of single-molecule tracks of HT-PBP2B across four septal diameters, plotted in polar coordinates. Tracks were produced using TrackMate[36] with 0.5 μm linking distance and 0 frame gaps, then plotted in polar coordinates using bespoke MATLAB code. All processive and immobile tracks are shown in blue, while diffusive tracks are shown in red. **(c)** Example radial kymograph of HT-PBP2B (rich media, 30 °C) showing a processive track and a long-lived diffusive track. As the diffusive track covered most of the cell circumference, two revolutions around the cell (0°–360° and 360°–720°) are plotted top and bottom, separated by a dotted yellow line. **(d)** Example mean-squared displacement (MSD) vs. time step plot for the kymograph shown in panel c. Blue circles: MSDs from processive track. Red circles: MSDs from diffusive track. Blue line: fit to MSDs from processive track. Red line: fit to MSDs from diffusive track (fit details in Methods). **(e)** Histogram of effective diffusion coefficients from MSD analyses. **(f)** Histogram of anomalous diffusion exponents $\alpha$ from MSD analyses. Sample sizes are listed in Supplementary Table 6.

**a**

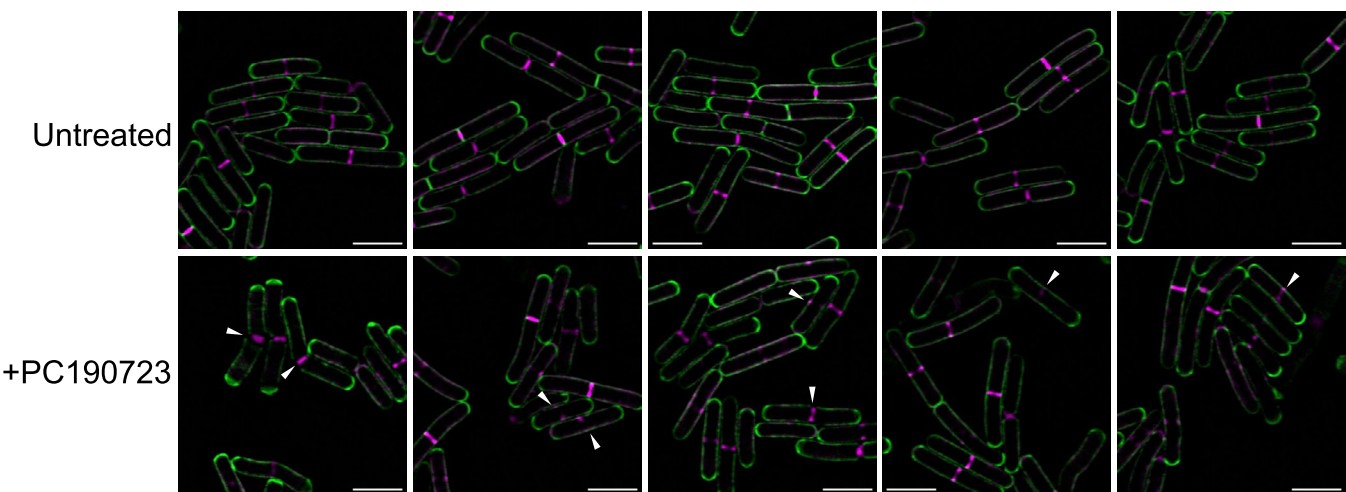

**b**

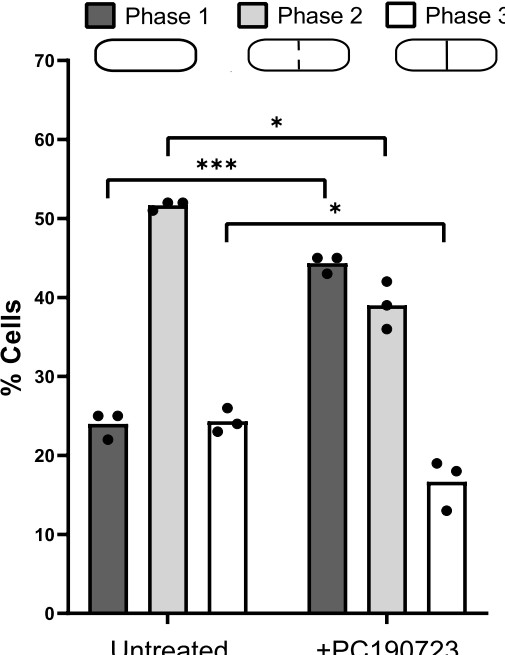

**Extended Data Fig. 5 | FtsZ treadmilling is not required for septal PG synthesis.** Cells (strain PY79; Supplementary Table 1) were stained with the fluorescent D-amino acid BADA for 90 min, then with TADA for 10 min. To arrest treadmilling by FtsZ, cells were treated with 14 μM PC190723 (5×MIC) for 10 min during the TADA staining step. **(a)** Panels show representative images of stained cells (green: BADA; magenta: TADA). White arrows show septal aberrations. Scale bars: 3 μm. **(b)** Quantification of the percentage of cells in each division phase in cells stained with BADA and TADA with or without PC190723. At least 100 cells were counted per repeat. Sample sizes are listed in Supplementary Table 6. The histograms show the means, and $p$-values are a result of unpaired $t$-tests with Welch's correction (two-tailed). ***: $p = 0.0002$ (phase 1), *: $p = 0.0155$ (phase 2), $p = 0.0364$ (phase 3).

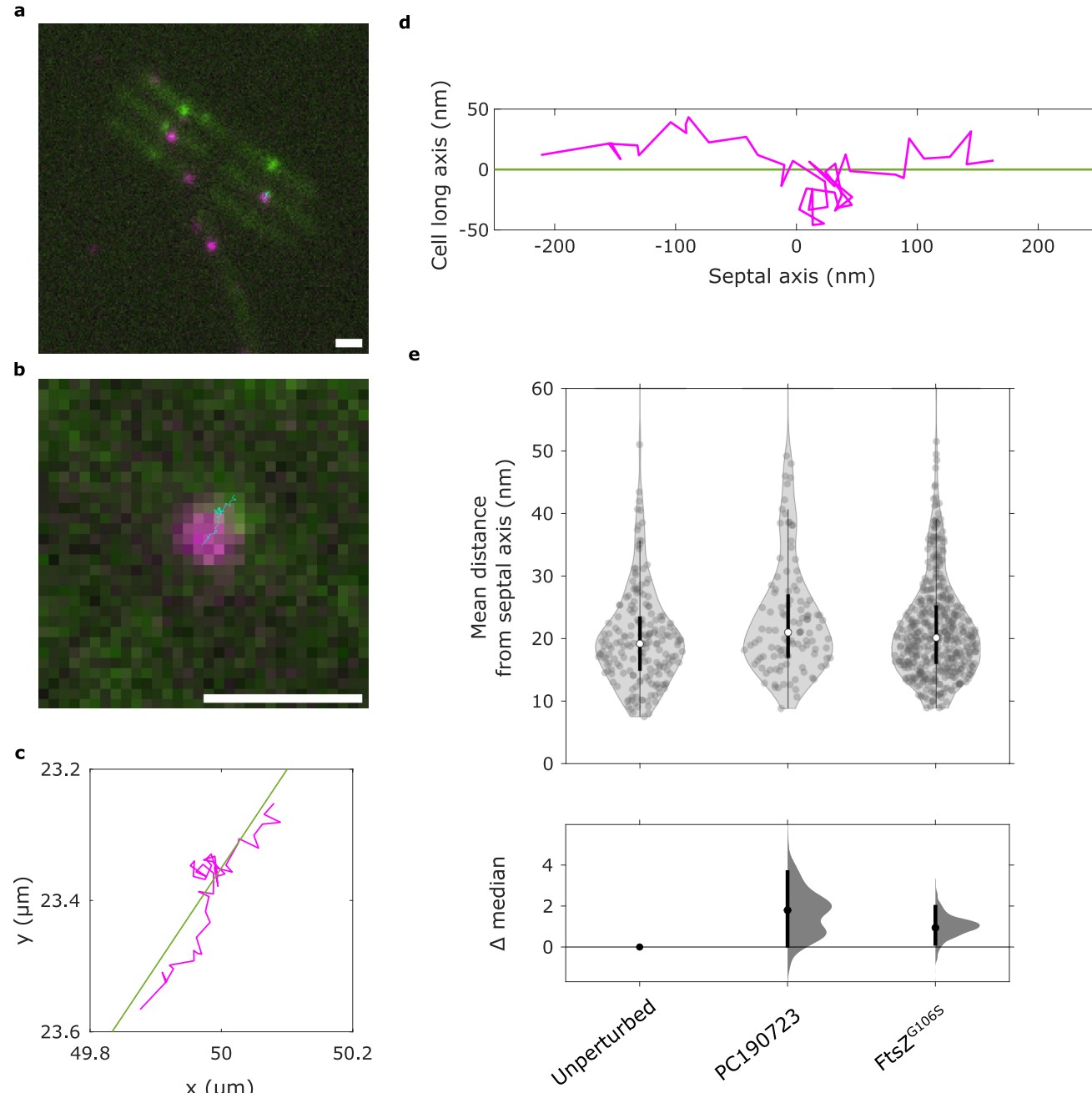

**Extended Data Fig. 6 | Motion of HT-PBP2B along the septal axis in horizontally-oriented cells. (a)** Example image of HT-PBP2B GFP-FtsZ Δhag strain (strain SH147; Supplementary Table 1) from a single-particle tracking video acquired with TIRF illumination (rich media, 30 °C). Shown is a merge of channels for imaging GFP-FtsZ (green) and HT-PBP2B (magenta). A single-molecule track is shown in cyan. Tracks were produced using TrackMate[36] with 0.1 μm linking distance and 0 frame gaps. Processive molecules were selected using the filters: >9 spots in track, >90 nm track displacement, <60 nm/s median track speed. **(b)** Zoom-in of single-molecule track of HT-PBP2B from panel **a** along a division septum. **(c)** Coordinates of the example track (magenta) along with a line showing the septal axis (green). **(d)** Example track from panel **c** rotated to show coordinates along the septal axis. **(e)** Mean distances of HT-PBP2B localizations from septal axes across conditions affecting FtsZ treadmilling. Each point represents the mean distance from the septal axis from a single processive HT-PBP2B track (for example the track in panel **d** represents one point). Violin plots: white circles, median; thick black lines, interquartile range; thin black lines, 1.5x interquartile range. DABEST plots: black circle, median difference between indicated conditions; black lines, 95% confidence interval of median difference. Microscope acquisition parameters are listed in Supplementary Table 5. Sample sizes are listed in Supplementary Table 6. Scale bars: 1 μm.

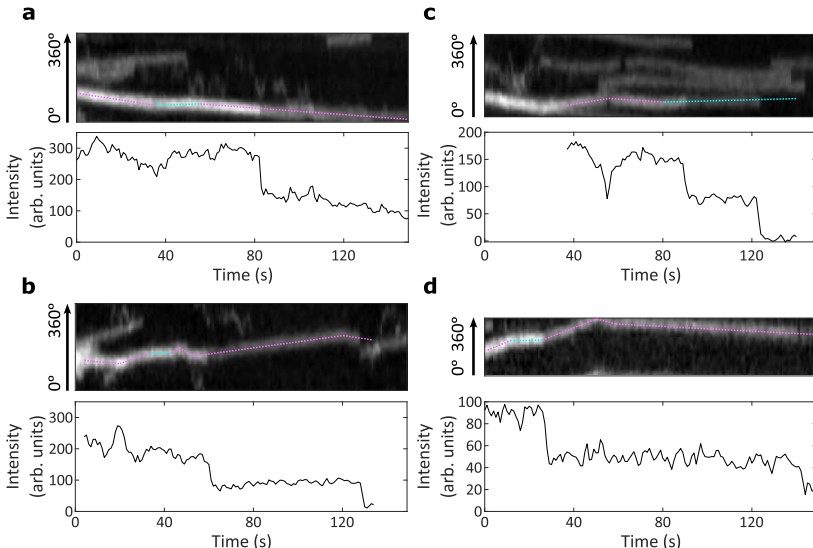

**Extended Data Fig. 7 | Fluorescence intensity drops during motion and after direction changes.** (**a**–**d**) Four example kymographs and accompanying intensity plots are shown. *Top:* Example kymographs of HT-PBP2B motion from smVerCINI videos showing discrete drops in fluorescence intensity during motion or after direction changes. Magenta segments: processive motion. Cyan segments: immobile. *Bottom:* Intensity traces for the tracks designated by dotted lines overlaid on kymographs.

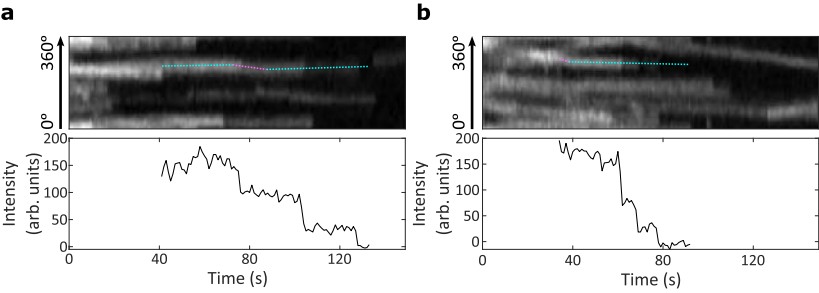

**Extended Data Fig. 8 | Multiple fluorescence intensity drops from single spots.** (**a**,**b**) Two example kymographs and accompanying intensity plots are shown. *Top:* Example kymographs of HT-PBP2B motion from smVerCINI videos showing several discrete drops in fluorescence intensity. Magenta segments: processive motion. Cyan segments: immobile. *Bottom:* Intensity traces for the tracks designated by dotted lines overlaid on kymographs.

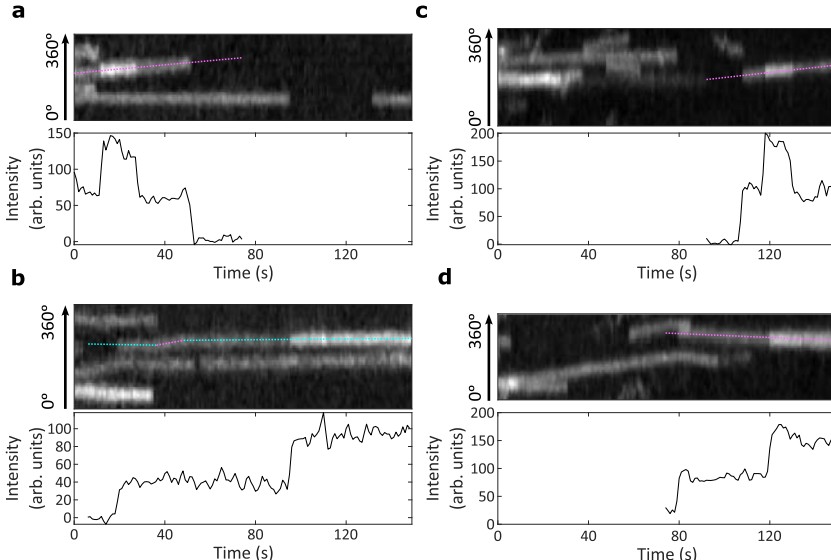

**Extended Data Fig. 9 | Fluorescence intensity jumps. (a–d)** Four example kymographs and accompanying intensity plots are shown. *Top:* Example kymographs of HT-PBP2B motion from smVerCINI videos showing discrete jumps in fluorescence intensity. Magenta segments: processive motion. Cyan segments: immobile. *Bottom:* Intensity traces for the tracks designated by dotted lines overlaid on kymographs.

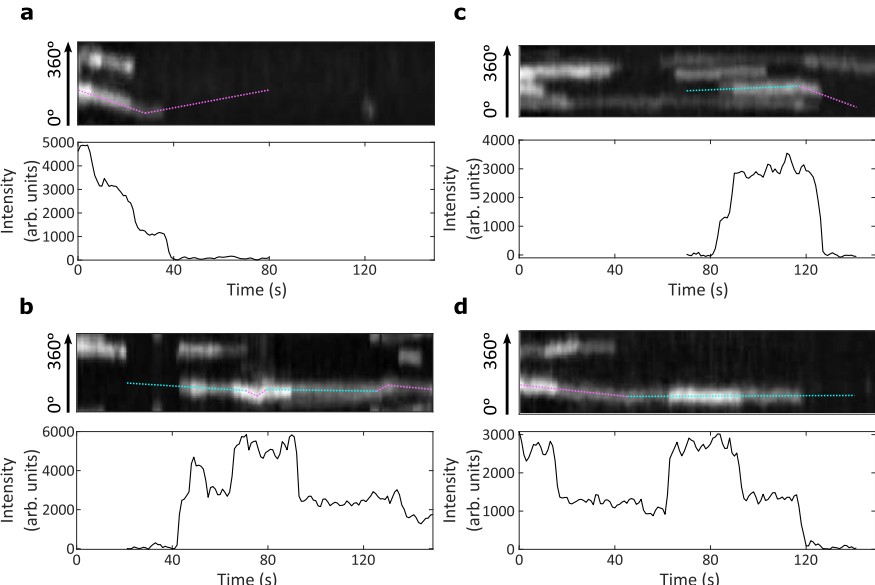

**Extended Data Fig. 10 | Fluorescence intensity drops and jumps in HT-FtsW tracks. (a–d)** Four example kymographs and accompanying intensity plots are shown for tracks of JFX554-HT-FtsW (strain bAB350; Supplementary Table 1). Cells were incubated with 5 nM JFX554 HT ligand for 15 min prior to imaging.

*Top:* Example kymographs of HT-FtsW motion from smVerCINI videos showing discrete drops or jumps in fluorescence intensity. Magenta segments: processive motion. Cyan segments: immobile. *Bottom:* Intensity traces for the tracks designated by dotted lines overlaid on kymographs.

# Reporting Summary

## Statistics

For all statistical analyses, confirm that the following items are present in the figure legend, table legend, main text, or Methods section.

| n/a | Confirmed | |
|---|---|---|
| ☐ | ☒ | The exact sample size (*n*) for each experimental group/condition, given as a discrete number and unit of measurement |
| ☐ | ☒ | A statement on whether measurements were taken from distinct samples or whether the same sample was measured repeatedly |
| ☐ | ☒ | The statistical test(s) used AND whether they are one- or two-sided *Only common tests should be described solely by name; describe more complex techniques in the Methods section.* |
| ☒ | ☐ | A description of all covariates tested |
| ☐ | ☒ | A description of any assumptions or corrections, such as tests of normality and adjustment for multiple comparisons |
| ☐ | ☒ | A full description of the statistical parameters including central tendency (e.g. means) or other basic estimates (e.g. regression coefficient) AND variation (e.g. standard deviation) or associated estimates of uncertainty (e.g. confidence intervals) |
| ☐ | ☒ | For null hypothesis testing, the test statistic (e.g. *F*, *t*, *r*) with confidence intervals, effect sizes, degrees of freedom and *P* value noted *Give P values as exact values whenever suitable.* |
| ☒ | ☐ | For Bayesian analysis, information on the choice of priors and Markov chain Monte Carlo settings |
| ☒ | ☐ | For hierarchical and complex designs, identification of the appropriate level for tests and full reporting of outcomes |
| ☒ | ☐ | Estimates of effect sizes (e.g. Cohen's *d*, Pearson's *r*), indicating how they were calculated |

*Our web collection on statistics for biologists contains articles on many of the points above.*

## Software and code

Policy information about availability of computer code

| Data collection | Data collected using either Micro-Manager (v2.0gamma), NS-Elements (v5.42.02), or Zen Black 2.3 (v16.0.14.316). |
|---|---|
| Data analysis | Videos analysed using Fiji (v1.54) with open-source plugins PureDenoise-GPU (v0.1.0) and PureDenoise-CPU (v0.1.0) (https://github.com/ZikaiSun/PureGpu/tree/main). |
| | Extended Data Figures 4 and 6, along with Supplementary Videos 14-16 show data from molecules manually tracked using TrackMate (v7.10.2). |
| | Further data analysis done using Matlab with custom code available on the Whitley lab Github page: https://github.com/WhitleyLab/Vercini_spt_analysis |
| | The divisome complex was modelled by AlphaFold2, using ColabFold (v1.3.0) and AlphaFold-Multimer (v2). |

For manuscripts utilizing custom algorithms or software that are central to the research but not yet described in published literature, software must be made available to editors and reviewers. We strongly encourage code deposition in a community repository (e.g. GitHub). See the Nature Portfolio guidelines for submitting code & software for further information.

## Data

Policy information about availability of data

All manuscripts must include a data availability statement. This statement should provide the following information, where applicable:

- Accession codes, unique identifiers, or web links for publicly available datasets
- A description of any restrictions on data availability
- For clinical datasets or third party data, please ensure that the statement adheres to our policy

All source data for figures and results in this paper can be found in the Figshare repository: https://doi.org/10.25405/data.ncl.c.7078312

The sequences for performing protein structure predictions were downloaded from the UniProtKB database (Q07868 (PBP2B); Q07867 (FtsL); O07639 (FtsW); P16655 (DivIB); P37471 (DivIC)).

## Research involving human participants, their data, or biological material

Policy information about studies with human participants or human data. See also policy information about sex, gender (identity/presentation), and sexual orientation and race, ethnicity and racism.

| Reporting on sex and gender | N/A |
|---|---|
| Reporting on race, ethnicity, or other socially relevant groupings | N/A |
| Population characteristics | N/A |
| Recruitment | N/A |
| Ethics oversight | N/A |

Note that full information on the approval of the study protocol must also be provided in the manuscript.

# Field-specific reporting

Please select the one below that is the best fit for your research. If you are not sure, read the appropriate sections before making your selection.

☒ Life sciences ☐ Behavioural & social sciences ☐ Ecological, evolutionary & environmental sciences

For a reference copy of the document with all sections, see nature.com/documents/nr-reporting-summary-flat.pdf

# Life sciences study design

All studies must disclose on these points even when the disclosure is negative.

| Sample size | No a priori sample size calculations were performed. No specific sample size was chosen (exceptions noted below) as the single cell/ single molecule nature of the measurements means moderate to large sample size, sufficient for robust statistical analysis, is usually straightforward to achieve. N>80 for each experiment or analysis in the paper, which was sufficient to evaluate results. Where the data were suitable, sample data violin plots/ histograms were evaluated post-hoc to check that the probability density function of the underlying distribution appeared well sampled; we observed that this was the case for all measurements. Numbers of cells, track segments, and other data points are all listed in Supplementary Table 6. <br> For bulk bacterial growth curves (Supplementary Figures 3 and 11b), three samples were prepared in order to estimate the variance of the measurement. |
|---|---|
| Data exclusions | No data was excluded. |
| Replication | The number of independent biological replicates for each experiment, defined as the number of experiments done using distinct and independently-prepared samples, can be found in Supplementary Table 6. |
| Randomization | Allocating experimental groups was not relevant for this study, as all bacterial cells of a particular strain are genetic clones. |
| Blinding | Blinding was neither possible nor necessary for this study, as 1) all bacterial cells of a particular strain are genetic clones and 2) analyses were not sufficiently subjective to require researcher blinding. |

# Reporting for specific materials, systems and methods

We require information from authors about some types of materials, experimental systems and methods used in many studies. Here, indicate whether each material, system or method listed is relevant to your study. If you are not sure if a list item applies to your research, read the appropriate section before selecting a response.

## Materials & experimental systems

| n/a | Involved in the study |
|---|---|
| ☐ | ☒ Antibodies |
| ☒ | ☐ Eukaryotic cell lines |
| ☒ | ☐ Palaeontology and archaeology |
| ☒ | ☐ Animals and other organisms |
| ☒ | ☐ Clinical data |
| ☒ | ☐ Dual use research of concern |
| ☒ | ☐ Plants |

## Methods

| n/a | Involved in the study |
|---|---|
| ☒ | ☐ ChIP-seq |
| ☒ | ☐ Flow cytometry |
| ☒ | ☐ MRI-based neuroimaging |

## Antibodies

| Antibodies used | Anti-Pbp2B, Merck, Cat. No. ABS2199<br>Anti-Spo0J, gift from Jeffery Errington lab<br>HRP-conjugated Anti-Rabbit IgG, Merck, Cat. No. A8275 |
|---|---|
| Validation | Anti-Pbp2B (from Merck website):<br>Application Anti-Pbp2B, Cat. No. ABS2199, is a rabbit polyclonal antibody that detects Penicillin-binding protein 2B (Pbp2B) and has been tested for use in immunofluorescence and Western Blotting.<br>Western Blotting Analysis: A 1:10,000 dilution from a representative lot detected Pbp2B in WT Bacillus Subtilis and GFP-Pbp2B (Courtesy of Dr. Richard Daniel at Newcastle University, UK).<br>Western Blotting Analysis: A 1:10,000 dilution from a representative lot detected His-PbpB recombinant protein (Courtesy of Dr. Richard Daniel at Newcastle University, UK).<br>Immunofluorescence Analysis: A representative lot detected Pbp2B in immunofluorescence applications (Daniel, R.A., et. al. (2000). Mol Microbiol. 35(2):299-311).<br>Western Blotting Analysis: A representative lot detected Pbp2B in Western Blotting applications (Bisson-Filho, A.W., et. al. (2017). Science. 355(6326):739-743; Adams, D.W., et. al. (2016). Mol Microbiol. 99(6):1028-42).<br><br>Anti-Spo0J (non-commercial):<br>Anti-Spo0J is a rabbit polyclonal antibody that detects Spo0J and has been tested for use in immunofluorescence and Western blotting: Glaser, P. & Sharpe, M.E. et al. (1997). Genes and Development. 11(9):1160-8.<br><br>Anti-Rabbit IgG (from Merck website):<br>Co-immunoprecipation and western blot analysis of C33A cell lysates were performed using HRP conjugated goat anti-rabbit IgG as the secondary antibody.<br>Immunohistochemistry was performed on frozen sections (10um) of mouse intestine, liver, and spleen using HRP-conjugated goat anti-rabbit IgG as the secondary antibody. Prior to incubation with the secondary, sections were treated with a mixture of MeOH/hydrogen peroxide 30% to block endogenouse peroxidases.<br>Prepared using the periodate method described by Wilson, M.B., and Nakane, P.K., in Immunofluorescence and Related Staining Techniques, Elsevier/North Holland Biomedical Press, Amsterdam, p215 (1978). |

