## [Peer Review File · Nature Microbiology]

Peer Review Information

Journal: Nature Microbiology

Manuscript Title: Peptidoglycan synthesis drives a single population of septal cell wall synthases during division in *Bacillus subtilis*

Corresponding author name(s): Dr Kevin Whitley

Reviewer Comments & Decisions:Decision Letter, initial version:

Message: 5th September 2023

Dear Dr Whitley,

Thank you for your patience while your manuscript "A one-track model for spatiotemporal coordination of *Bacillus subtilis* septal cell wall synthesis" was under peer-review at Nature Microbiology. It has now been seen by 5 referees, whose expertise and comments you will find at the end of this email. Although they find your work of some potential interest, they have raised a number of concerns that will need to be addressed before we can consider publication of the work in Nature Microbiology.

In particular, you will see that several of the referees highlighted that a number of additional controls and experimental analyses including reanalysis of the FtsZ(G106S) mutant strain and use of antibiotics or mutant constructs to disrupt peptidoglycan synthesis are needed to strengthen support for the conclusion that PBP2B movement is independent of FtsZ activity. Furthermore, there were also concerns that further data were needed to support the conclusions that PBP2B undergoes multimerisation, and that FtsZ plays a regulatory role in maintaining efficient peptidoglycan synthesis. Several referees were also surprised by the lack of effect of temperature upon FtsZ activity, and requested further information or validation of certain strains and constructs used in the analyses. We feel that these are critical points which would need to be addressed for us to further consider a revised manuscript, alongside the remaining issues outlined in the referees' reports, which are clear and should be straightforward to address.

Should further experimental data allow you to address these criticisms, we would be happy to look at a revised manuscript.

Please include a data availability statement as a separate section after Methods but before references, under the heading "Data Availability". This section should inform readers about the availability of the data used to support the conclusions of your study. This information includes accession codes to public repositories (data banks for protein, DNA or

3RNA sequences, microarray, proteomics data etc...), references to source data published alongside the paper, unique identifiers such as URLs to data repository entries, or data set DOIs, and any other statement about data availability. At a minimum, you should include the following statement: "The data that support the findings of this study are available from the corresponding author upon request", mentioning any restrictions on availability. If DOIs are provided, we also strongly encourage including these in the Reference list (authors, title, publisher (repository name), identifier, year). For more guidance on how to write this section please see: <http://www.nature.com/authors/policies/data/data-availability-statements-data-citations.pdf>

* If you have not done so already we suggest that you begin to revise your manuscript so that it conforms to our Article format instructions at <http://www.nature.com/nmicrobiol/info/final-submission>. Refer also to any guidelines provided in this letter.

When submitting the revised version of your manuscript, please pay close attention to our [href="https://www.nature.com/nature-portfolio/editorial-policies/image-integrity">Digital Image Integrity Guidelines](https://www.nature.com/nature-portfolio/editorial-policies/image-integrity). and to the following points below:

Note: This url links to your confidential homepage and associated information about manuscripts you may have submitted or be reviewing for us. If you wish to forward this e-mail to co-authors, please delete this link to your homepage first.

Nature Microbiology is committed to improving transparency in authorship. As part of our

efforts in this direction, we are now requesting that all authors identified as 'corresponding author' on published papers create and link their Open Researcher and Contributor Identifier (ORCID) with their account on the Manuscript Tracking System (MTS), prior to acceptance. This applies to primary research papers only. ORCID helps the scientific community achieve unambiguous attribution of all scholarly contributions. You can create and link your ORCID from the home page of the MTS by clicking on 'Modify my Springer Nature account'. For more information please visit please visit www.springernature.com/orcid.

If you wish to submit a suitably revised manuscript we would hope to receive it within 6 months. If you cannot send it within this time, please let us know. We will be happy to consider your revision, even if a similar study has been accepted for publication at Nature Microbiology or published elsewhere (up to a maximum of 6 months).

Yours sincerely,

Reviewer Expertise:

Referee #1: Cell biology/cell division, B. subtilis biology
Referee #2: cell wall synthesis, Gram positive cell biology
Referee #3: Cell division Gram positives

Reviewer Comments:

Reviewer #1 (Remarks to the Author):

Bacterial division is initiated by assembly of the tubulin-like protein FtsZ at the future division site. FtsZ forms short, treadmilling polymers that recruit components of the cell wall synthesis machinery. Initial work suggested treadmilling was important for guiding cell wall synthesis. In *Escherichia coli* the current model suggests peptidoglycan synthases are divided into two populations: an active population uncoupled from FtsZ treadmilling and an inactive population that is recruited around the nascent division site by treadmilling FtsZ polymers, to ensure the division machinery is evenly distributed across the division plane. This two-track model has been corroborated in *Caulobacter crescentus*, but not in Gram positive organisms. In particular, previous work in *Streptococcus pneumoniae* indicates that PG synthesis enzymes move independently of FtsZ treadmilling (Perez, 2019 PMID: 30718427). Additionally, work in *S. aureus* suggests that PG synthesis is also independent of FtsZ-treadmilling, particularly once cell wall synthesis becomes focused on building a septum (Monteiro, 2018 PMID 29443967).

Whitley et al, together with the parallel submission from Schaper and colleagues, provide additional data arguing against a two-track FtsZ treadmilling dependent model in *Bacillus subtilis* and *Staphylococcus aureus*, suggest the firmicutes. Instead, there appears to be only a single population of peptidoglycan synthesis enzymes which move processively independent of FtsZ treadmilling. At the same time, FtsZ treadmilling does contribute to ensuring efficient PG synthase activity and septum formation. Based on these observations the authors propose that FtsZ acts as a guide to prevent PG synthases from wandering away from the septal plane, thereby facilitating PG synthesis along a narrow

axis. This last point is purely speculative, given the absence of data indicating coordination between FtsZ and the PG synthesis machinery. Methodologically, the experiments are well conceived and necessary experimental controls are present. All conclusions are convincingly supported by the data and statistical tests appear to be appropriate. The AlphaFold model is supported by Käshammer et al. 2023 (as the authors themselves note).

Comments:

1. The authors need to make it clear in the introduction and conclusion that Perez et al and Montiero et al previously identified disconnects between FtsZ treadmilling and PG synthesis in *S. pneumoniae* and *S. aureus* respectively in the introduction. While these studies did not look at *B. subtilis*, they suggest that the relationship between FtsZ treadmilling and PG synthesis is different in the firmicutes. Although both studies are mentioned in the introduction, this point is not as clear as it should be.
2. Please include FtsZ speed was included in 2c to emphasize that it does not correlate with constriction rate.
3. How often were jumps in fluorescence intensity observed? In most of the data? Half? 10%? Less? Did observed intensity changes cooccur with any changes in PBP2B speed?
4. To fully make the case that treadmilling is not required for PG cross wall synthesis, the authors should include data (TEM or similar) demonstrating the quality of the septum in mutant cells and/or post antibiotic treatment.
5. Is there sufficient resolution to visualize PBP2B "wobbling" along the z ring if the cells are imaged perpendicular to the viewing plane or is that not feasible?
6. Do micro well sizes used in smVerCINI exclude a substantial portion of the cell population? Is it possible that the excluded population may have different PBP2B or FtsZ dynamics?

Reviewer #2 (Remarks to the Author):

This is an interesting, timely paper about movement of the septal PG synthase in dividing cells of *B. subtilis*. The authors use a microhole approach to vertically orient *B. subtilis* cells that are expressing a low level of GFP-FtsZ to mark septal rings and HT-PBP2B, the Class B transpeptidase that interacts with the FtsW glycosyltransferase to form the septal PG synthase. The authors then track and analyze the motion HT-PBP2B under several different conditions. They come to a couple of very convincing major conclusions and one conclusion that needs further qualification because of alternative explanations. They show that HT-PBP2B moves processively driven by PG synthesis and not by FtsZ treadmilling. In addition, this work reveals non-motile and subdiffusive HT-PBP2B in the septal ring. They further show that processive movement of HT-PBP2B is asynchronous with FtsZ movement. In this analysis, they determine HT-PBP2B and FtsZ movement at different temperatures and in rich and poor media. This leads to an interesting set of physiologically relevant data showing that HT-PBP2B velocity increases with cell doubling time, consistent with PG synthesis driving movement and that septal constriction rates measured previously correlate with HT-PBP2B speed. The last major conclusion shows that HT-PBP2B intensity in tracks drops or increases by 2-, 3-, or even 4-fold, especially upon directional changes. This result is consistent with loading and unloading of sPG multimers to complexes. The one conclusion that is not well supported and is overstated throughout the manuscript is that processive motion "depends" on FtsZ treadmilling. There are a couple issues with this strong interpretation of a small drop in HT-PBP2B processive velocity in cells in which FtsZ GTPase and treadmilling are inhibited. These

concerns are discussed below in point 12.

The data are of high quality in this well-presented paper that will be of widespread interest to the field, because it corrects a previous conclusion about septal PG synthase movement in *B. subtilis*. Nevertheless, several issues and questions are raised by the data that need to be addressed. These points and suggestions are listed below.

1. Line one and throughout. The terminology "track" has been introduced in other papers in the field. However, the terminology becomes confusing in the model in Figure 5, where the "single-track" is the septal ring that consists of multiple separate tracks created by individual FtsZ filaments (Fig. 5a). The authors should consider labeling these multiple separate tracks as "sub-tracks" or "lanes" associate with separate FtsZ filaments to distinguish them from the idea of the larger "single-track" nomenclature.
2. Line 25, 58, and throughout. As discussed below in point 12, the direct regulation of PG synthesis activity by FtsZ treadmilling is not firmly established in this paper and should not be stated as such a strong conclusion.
3. Line 72 and figures in general. After "induction level" please list the optimal IPTG concentration used. In general, there are other places where information can be included in figures and text so that readers do not need to flip around in the paper. For example, please go through and include the simplified genotypes for the strains in the figures, rather than just the strain numbers. As written, the reader has to flip to the strain table or other places in the manuscript to get this information, which can easily be included in the figures.
4. Line 72 after "cell morphology". Is the cellular amount of HT-PBP2B known relative to the normal WT level, since HT-PBP2B is the sole source in these strains? Are we dealing with HT-PBP2B at normal WT levels or is HT-PBP2B over- or underexpressed in these experiments? If different from WT, how would HT-PBP2B impact interpretations?
5. Line 73. Is the GFP-FtsZ active? Does it perturb WT FtsZ function at all? Please clarify.
6. Fig. 1. The green GFP-FtsZ ring is used to find the plane for tracking. How thick is this ring relative to the resolution of this method? That is, if the plane of focus is moved up or down are new slices of tracks revealed. Please comment.
7. Fig. 1. A couple points about this figure. In general, there are additional interesting data and information here that are not brought out.
 - a. For general readers, the authors should indicate that diagonal tracks in Fig. 1c indicate processive motion, whereas horizontal tracks indicate lack of motion (Fig. 1d). The authors should also indicate that when a diagonal track changes direction as in Fig. 1c, it indicates change in direction of the HT-PBP2B molecules, including reversals.
 - b. It seems that most of the kymographs have two tracks, one diagonal (moving) and the other horizontal (non-moving), like Fig. 1c. Were kymographs detected that support the multiple sub-track arrangement shown in the model in Figure 5. If so, please include and highlight these.
 - c. What is the percentage of molecules that move processively without changing direction? How often do non-motile molecules start moving processively or processively moving molecule stop? Do processive or stopped molecules start moving diffusively?
 - d. What is the relative distribution of septal ring HT-PBP2b molecules that are moving processively, not moving, or diffusively moving. This interesting information is buried in

the frequencies in Fig. 1f-h, but is never pulled together.

e. It seem that the calculated lifetime of tracks include motion in more than one direction. Please specify this.

f. Are diffusing HT-PBP2B molecules detected outside of the septal ring area? If so, do these molecules show the higher velocity characteristic of free diffusion? Please comment.

8. Line 91. What is the basis for the assumption that the non-moving molecules are bound to the cell wall? How do you envision this binding? Incomplete TP half-reactions? As alternatives, the non-motile molecules may simply not be catalytically active or may be trapped in incomplete or aberrantly assembled complexes. Please qualify this speculation more.

9. In Fig. 1, Is the drop in speed of the sub-diffusing molecules in the presence of antibiotics significant, and if so, why does it happen?

10. Figure 2 raises a variety of important questions that need to be addressed or commented on.

a. Why does the FtsZ treadmilling speed not show a temperature dependence? The average treadmilling speed should be temperature dependent, because it depends on an enzymatic reaction. Please explain this.

b. Why are the distributions so different in rich medium at 37 degrees? Remarkably, the HT-PBP2B distribution pancakes out. Why is this? Why is the frequency of FtsZ treadmilling apparently much lower at 37 degrees in rich medium compared to the other three panels? Please offer some possible explanations for this singular behavior, which the text argues may have affected previous interpretations of data.

c. The FtsZ treadmilling is taken from previously reported experiments, presumably in different strains than the one expressing HT-PBP2b. If it is a different strain, was the treadmilling rate spot checked and could it be relevant to point b above? Please comment.

11. Fig. S5 and the correlations in Fig. 2a and 2c are particularly interesting and nice; also, the results showing multimeric complexes later.

12. Lines 174-198. There are several issues with the interpretation of the data presented in Figure 3. The data show that there is a moderate (about 40%) drop in HT-PBP2B processive speed when FtsZ treadmilling is inhibited partially (but enough to cause large morphology differences (Fig. S6)) or completely (PC19 compound). These data are convincing and correct the previous result that indicated no movement in the presence of the PC19 compound.

However, the conclusion that the slower HT-PBP2B velocity is a direct effect ("depends on") FtsZ treadmilling is not justified, because the change could be an indirect effect.

a. In particular, the paper nicely shows that HT-PBP2B velocity depends on the rate of PG synthesis, which is dependent on cell metabolism (e.g., fosfomycin (Fig. 1e); and growth rate (Fig. 2b)). Limiting or eliminating FtsZ treadmilling is bound to be highly pleiotropic and to set off all kinds of stress responses, including ones that may limit PG precursor synthesis, which will reduce HT-PBP2B velocity.

b. In addition, if HT-PBP2B speed is regulated by FtsZ treadmilling, then why does partial limitation of FtsZ treadmilling show the same dependence as complete inhibition of FtsZ treadmilling? It seems that the proposed regulation would be graded.

c. Last, why does the distribution of processively moving HT-PBP2B shift to lower speeds when FtsZ is partially inhibited in the mutant compared to completely inhibited by the PC19 compound?

The idea that treadmilling somehow is regulating HT-PBP2B speed directly as stated in the Abstract, here, the Discussion, and Figure 5 is not strongly supported by this single set of data in stressed cells. These sections should be rephrased and revised to include alternative interpretations.

13. Line 236. Was there a marked increase in non-linear HT-PBP2B movement detected in the FtsZ-G106S or PC19 experiments? The tracks shown in Fig. 3 do not seem to be different from those in Fig. 1c and elsewhere. In mutants of FtsZ filament bundling proteins, is HT-PBP2B velocity slower? This seems like a corollary of this model that could be tested; although again, more severe bundling mutants may be pleiotropic on metabolism and PG synthesis.

14. Line 263. Has the TIRFm experiment been repeated with these strains and conditions? Wasn't the motion determined previously in cells at different stages of septation and not just in cells at late stages of septation? For a population that contains lots of early divisional cells, why would the previously TIRFm measurement be biased and show no motion at all? This does not seem like a strong argument. Please consider revising or omitting.

15. Line 275. The examples presented indicate major differences and not "minor" differences as stated. Please change to: "there may remain differences". Also, please see point 12 above, supporting further qualification of the model presented throughout.

16. Line 287. Please add: HT-PBP2B in the septal ring (please see point 7f, above).

17. Line 423. Remove: "in some cases", since all of the diagrams seem to use this format.

18. Fig. S5. What are the times on the kymographs?

19. Fig. S9. A lot of these tracks look horizontal indicating non-motile. It would be helpful to indicate in all of the kymographs which tracks are considered to represent processively moving or non-motile molecules.

Reviewer #3 (Remarks to the Author):

The manuscript by Whitley et al. analyzes the dynamics of peptidoglycan synthesis during septation in *Bacillus subtilis*. By measuring PBP2B dynamics, the authors report that PBP2B moves at speed that is different than FtsZ treadmilling speed. The authors propose that the motion of PBP2B is mediated by peptidoglycan synthesis, and not by FtsZ treadmilling. Since they observed that the speed of PBP2B was affected when FtsZ treadmilling was inhibited, they conclude that PBP2B speed is only partially dependent on FtsZ treadmilling. The study proposes an evolved model for regulation of peptidoglycan synthesis during cell division in Gram-positive bacteria in which cell wall synthesis proceeds via a single population of a multimeric complex. The study certainly addresses an important question that should be of broad interest to bacterial cell biologists. However, I have a several experimental suggestions for additional controls that may strengthen the authors conclusions.

Major comments:

1. Figure S3. The authors characterize strain SH147 (inducible FtsZ-sGFP and HALO-pbp2B) to show that the growth curves are largely unaffected by induction of both constructs. Please include micrographs and analysis of cell lengths to see if cell morphology is affected by induction of these constructs. The subsequent data in the paper may still be valid even if morphology is affected, but the reader should be aware of the physiological consequences for the cell of producing these molecules.
2. Fig. 2. The presented results for speeds of PBP2B and FtsZ are consistent with the authors' model of PG synthesis being largely independent on FtsZ treadmilling. That said, since FtsZ treadmilling depends on GTPase activity, and I would assume that GTPase activity of FtsZ is dependent on temperature, do the authors have a sense of why FtsZ treadmilling speed does not appreciably vary across the different conditions tested? Do the authors have a different (perhaps genetic) method of varying FtsZ treadmilling speed to demonstrate that PBP2B speed is unaffected under that condition?
4. A major question that is not addressed by the manuscript is "what drives PBP2B movement in the septum?". One possibility is the enzymatic activity of the complex due to the availability of substrates or influence of temperature. Can the authors analyze the dynamics of PBP2B in the presence of antibiotics that target PBP2B activity or test the behavior of PBP2B variants that display reduced enzymatic activity?
4. Fig. 3e. The authors report processive PBP2B speed as a function of FtsZ treadmilling speed, but it is unclear if the authors independently measured the treadmilling speed of the G106S variant in this study or if they simply used a previously published result. Since FtsZ treadmilling is a central part of the author's argument, please formally measure the treadmilling speed of G106S, in this study if it was not done already.
5. The authors' conclusion that the divisome synthesis complex is multimeric is based solely on the fluorescence intensity plots in Fig. 4a. If the machinery that synthesizes septal PG is indeed multimeric, would the authors be able to measure similar behavior with another member of the complex (for example, FtsW), or via a dual labeling experiment with PBP2B and FtsW?

Author Rebuttal to Initial comments

Author response:

We would like to thank all the reviewers for their helpful comments and for their careful reading of the manuscript. We have collected new data, incorporated new analysis into the text, and revised the manuscript to address all the concerns raised by the reviewers, including all of the points highlighted in the editor's comments.

Below we respond to the criticisms raised by the reviewers point by point. We have noted **in bold** changes made to the manuscript.

Reviewer Expertise:

10Referee #1: Cell biology/cell division, B. subtilis biology

Referee #2: cell wall synthesis, Gram positive cell biology

Referee #3: Cell division Gram positives

Reviewer Comments:

Reviewer #1 (Remarks to the Author):

Bacterial division is initiated by assembly of the tubulin-like protein FtsZ at the future division site. FtsZ forms short, treadmilling polymers that recruit components of the cell wall synthesis machinery. Initial work suggested treadmilling was important for guiding cell wall synthesis. In Escherichia coli the current model suggests peptidoglycan synthases are divided into two populations: an active population uncoupled from FtsZ treadmilling and an inactive population that is recruited around the nascent division site by treadmilling FtsZ polymers, to ensure the division machinery is evenly distributed across the division plane. This two-track model has been corroborated in Caulobacter crescentus, but not in Gram positive organisms. In particular, previous work in Streptococcus pneumoniae indicates that PG synthesis enzymes move independently of FtsZ treadmilling (Perez, 2019 PMID: 30718427). Additionally, work in S. aureus suggests that PG synthesis is also independent of FtsZ-treadmilling, particularly once cell wall synthesis becomes focused on building a septum (Monteiro, 2018 PMID 29443967).

Whitley et al, together with the parallel submission from Schaper and colleagues, provide additional data arguing against a two-track FtsZ treadmilling dependent model in Bacillus subtilis and Staphylococcus aureus, suggest the firmicutes. Instead, there appears to be only a single population of peptidoglycan synthesis enzymes which move processively independent of FtsZ treadmilling. At the same time, FtsZ treadmilling does contribute to ensuring efficient PG synthase activity and septum formation. Based on these observations the authors propose that FtsZ acts as a guide to prevent PG synthases from wandering away from the septal plane, thereby facilitating PG synthesis along a narrow axis. This last point is purely speculative, given the absence of data indicating coordination between FtsZ and the PG synthesis machinery.

Methodologically, the experiments are well conceived and necessary experimental controls are present. All conclusions are convincingly supported by the data and statistical tests appear to

be appropriate. The AlphaFold model is supported by Käshammer et al. 2023 (as the authors themselves note).

We thank the Reviewer for their positive comments and their insightful suggestions. We have now carefully addressed the points raised. In particular, we have performed further experiments to test the model in our first submission of FtsZ acting as a guide to direct motion of the divisome synthesis complex. This is elaborated below in our response to Comment 5.

Comments:

1. The authors need to make it clear in the introduction and conclusion that Perez et al and Montiero et al previously identified disconnects between FtsZ treadmilling and PG synthesis in *S. pneumococcus* and *S. aureus* respectively in the introduction. While these studies did not look at *B. subtilis*, they suggest that the relationship between FtsZ treadmilling and PG synthesis is different in the firmicutes. Although both studies are mentioned in the introduction, this point is not as clear as it should be.

We have now modified the Introduction and Discussion to make this point clearer. Changes to the Introduction (Lines 41-54) are reproduced here:

Several models have been proposed to explain how these proteins cooperate to enact division despite such different motion patterns⁸. We previously proposed a model where FtsZ treadmilling drives septal constriction in *B. subtilis* as a coupled cytoskeleton-synthase complex². However, subsequent work in this and other Bacillota (a.k.a. Firmicute) species demonstrated that sPG synthesis is not tightly coupled to FtsZ treadmilling. FtsZ treadmilling is dispensable for septal constriction after constriction has initiated in both *Staphylococcus aureus* and *B. subtilis*^{4,9}, and the motions of divisome synthesis complexes are uncoupled from treadmilling FtsZ filaments in *Streptococcus pneumoniae*⁵. Meanwhile, work on the Pseudomonadota (a.k.a. Proteobacteria) species *Escherichia coli* and *Caulobacter crescentus* has supported a model where the synthase complex moves on two 'tracks': an FtsZ-track where inactive synthase complexes are distributed around the division septum and an sPG-track where synthase complexes build the cell wall independently of FtsZ¹⁰⁻¹³. According to this model, an activating protein (FtsN in *E. coli* and FzIA in *C. crescentus*) is required to initiate active sPG synthesis on the sPG-

track^{13,14}. However, it is unclear how far this model generalises across the bacterial domain, as many species lack a known activator of cell division.

2. Please include FtsZ speed was included in 2c to emphasize that it does not correlate with constriction rate.

We have now included FtsZ treadmilling speed in Figure 2c. For visual clarity and for better comparison with the data presented in Figure 2b, we have also reversed the axes.

3. How often were jumps in fluorescence intensity observed? In most of the data? Half? 10%? Less? Did observed intensity changes cooccur with any changes in PBP2B speed?

We have now quantified the frequency of fluorescence drops and jumps in our data and have added this to the text to strengthen our conclusion that the divisome synthesis complex is multimeric. We also discuss the frequency of these drops and jumps co-occurring with changes in PBP2B behavior. These changes can be found in Lines 275-293, reproduced here:

During this study, we observed many cases where the fluorescence intensities of HT-PBP2B spots showed discrete drops to half their value (Figure 4a, b; Supplementary Videos 2 and 11), indicating the presence of two copies of the fluorescently-labelled protein. Under our standard conditions (rich media, 30°C, 100 μ M IPTG, 250 pM JFX554 HaloTag ligand), we observed these intensity drops in 11% (N=56) of full tracks. We also observed multiple occasions where such intensity drops occurred during motion and even after direction changes (Figure 4a, b; Supplementary Figure 18), strongly suggesting that multiple monomers of HT-PBP2B are moving together as part of a larger complex. Due to the sub-stoichiometric nature of the labelling method, we cannot precisely quantify the number of HT-PBP2B molecules in a given complex, although we have observed rare cases with even three or four such drops in intensity (Supplementary Figure 19).

Surprisingly, we also observe cases where the fluorescence intensity signal shows discrete jumps to twice their value (4% (N=22) of full tracks under our standard conditions; Figure 4c, d; Supplementary Figure 20; Supplementary Videos 12 and 13). We observe these discrete jumps in both immobile and processive tracks. This suggests that the oligomeric state of synthesis complexes is dynamic, where new PBP2B molecules can bind to both active and inactive complexes. As only 3 out

of 22 observed intensity jumps (14%) and 9 out of 56 (16%) of observed intensity drops roughly corresponded to a change in HT-PBP2B speed, it appears that these events do not necessarily affect divisome synthesis complex activity, although it remains possible that there is a higher probability of speed change during an intensity drop/jump event than without.

4. To fully make the case that treadmilling is not required for PG cross wall synthesis, the authors should include data (TEM or similar) demonstrating the quality of the septum in mutant cells and/or post antibiotic treatment.

To investigate this point we treated cells with high levels of PC190723 to inhibit FtsZ treadmilling for approximately one round of cell division while simultaneously labelling with fluorescent D-amino acids to specifically identify septal PG synthesis which had occurred during antibiotic treatment. We imaged cells by SIM super-resolution microscopy to analyse septal morphology and progression through the cell division cycle. As expected, the number of cells without septa increased substantially, indicating that inhibition of FtsZ treadmilling inhibits cell division initiation as we reported previously (Whitley et al. *Nature Communications* 2021). For cells where we observed septation during antibiotic treatment – i.e. where constriction initiation had already begun prior to treatment – septa looked mostly similar to untreated cells, with only modest septation defects (highlighted with arrows in Supplementary Figure 14a), consistent with our conclusion that treadmilling is not required for PG cross-wall synthesis.

We have added these data as Supplementary Figure 14 and brief discussion of this point (Lines 239- 242), reproduced here:

We confirmed that sPG synthesis continues in the absence of FtsZ treadmilling by treating cells with PC190723 for approximately one round of cell division while simultaneously labelling them with fluorescent D-amino acids, then imaging with Structured Illumination Microscopy (Methods; Supplementary Figure 14).

5. Is there sufficient resolution to visualize PBP2B “wobbling” along the z ring if the cells are imaged perpendicular to the viewing plane or is than not feasible?

To test our model of a ‘broken track’ (depicted in Figure 5 of our first submission) we have now imaged the motion of HT-PBP2B along septa in horizontally-oriented cells and quantified the off-axis motion with and without perturbations to FtsZ treadmilling. Although we do measure a slight increase in off- axis localizations when FtsZ treadmilling is perturbed, the size of the increase is only ~2 nm, which is unlikely to be biologically meaningful. This suggests to us that the model we initially presented does not explain this reduction in speeds.

We now describe this new experiment in the text (Lines 252-262) and present the new data on the off-axis motion of HT-PBP2B as Supplementary Figure 17 and Supplementary Videos 14-16. We have also removed the description of our earlier model in the text of Discussion and replaced Figure 5 with a new model cartoon. We now include a new paragraph in Discussion (Lines 353-361) with a few possible reasons for the reduction in HT-PBP2B speeds we observe when FtsZ treadmilling is perturbed. Addition to the Results section (Lines 252-262) is reproduced here:

We recently reported evidence from computational studies that perturbations to FtsZ treadmilling disrupt the nematic order of the FtsZ filament network²². We therefore considered that the reduction in HT-PBP2B speeds could result from motion along a disordered, jagged path due to transient interactions between the divisome synthesis complex and randomly-oriented FtsAZ filaments. To test this, we imaged the processive motion of HT-PBP2B molecules in horizontally-oriented cells and measured the displacements from the septal axis (Methods; Supplementary Figure 17). With either PC190723 treatment or expression of FtsZ^{G106S}, the median off-axis displacements were within 2 nm of that measured for the unperturbed case, which is unlikely to be biologically meaningful (Supplementary

Figure 17e). This suggests that the reduction in HT-PBP2B speeds observed upon perturbations to FtsZ treadmilling does not result from HT-PBP2B off-axis motion.

Changes to the description of our model in the Discussion section are reproduced here (Lines 310- 315):

Our results show that a multimeric divisome synthesis complex in *B. subtilis* follows a single track dependent on sPG synthesis and asynchronous with FtsZ treadmilling. This sharply contrasts with two prominent models for bacterial cell division^{2,11}, which predict the existence of a processive population of synthesis complexes associated with FtsZ treadmilling. Our results instead support a model of septal PG synthesis where the Z-ring recruits the synthesis complex to the septal leading edge but does not directly regulate its motion and synthesis activity (Figure 5).

6. Do micro well sizes used in smVerCINI exclude a substantial portion of the cell population? Is it possible that the excluded population may have different PBP2B or FtsZ dynamics?

All micro-holes used here were 6.7 μm deep. As shown in Supplementary Figure 2c-d, the lengths of cells under our standard culturing and imaging conditions (rich media, 30C, 100 μM IPTG, 0.075% xylose) were 2.73 μm (IQR [2.02, 3.38]). This population includes cells across all stages of the cell cycle, from birth to division.

The widths of micro-holes used here varied from 1.0 to 1.3 μm , as this range was found to be ideal for trapping *B. subtilis* PY79 cells used here. Consistent with previous work showing the narrow width distribution for WT cells, we measure the widths under our standard conditions to be 0.78 μm (IQR [0.73, 0.81]).

Therefore, the micro-hole sizes we have used here are sufficient to trap the full population of cells undergoing division.

Reviewer #2 (Remarks to the Author):

*This is an interesting, timely paper about movement of the septal PG synthase in dividing cells of *B. subtilis*. The authors use a microhole approach to vertically orient *B. subtilis* cells that are expressing a low level of GFP-FtsZ to mark septal rings and HT-PBP2B, the Class B transpeptidase that interacts with the FtsW glycosyltransferase to form the septal PG synthase. The authors then track and analyze the motion HT-PBP2B under several different conditions. They come to a couple of very convincing major conclusions and one conclusion that needs further qualification because of alternative explanations. They show that HT-PBP2B moves processively driven by PG synthesis and not by FtsZ treadmilling. In addition, this work reveals non-motile and subdiffusive HT-PBP2B in the septal ring. They further show that processive movement of HT-PBP2B is asynchronous with FtsZ movement. In this analysis, they determine HT-PBP2B and FtsZ movement at different temperatures and in rich and poor media. This leads to an interesting set of physiologically relevant data showing that HT-PBP2B velocity increases with cell doubling time, consistent with PG synthesis driving movement and that septal constriction rates measured previously correlate with HT-PBP2B speed. The last major conclusion shows that HT-PBP2B intensity in tracks drops or increases by 2-, 3-, or even 4-fold, especially upon directional changes. This result is consistent with loading and unloading of sPG multimers to complexes. The one conclusion that is not well supported and is overstated throughout the manuscript is that processive motion “depends” on FtsZ treadmilling. There are a couple issues with this strong interpretation of a small drop in HT-PBP2B processive velocity in cells in which FtsZ GTPase and treadmilling are inhibited. These concerns are discussed below in point 12.*

*The data are of high quality in this well-presented paper that will be of widespread interest to the field, because it corrects a previous conclusion about septal PG synthase movement in *B. subtilis*. Nevertheless, several issues and questions are raised by the data that need to be addressed. These points and suggestions are listed below.*

We thank the Reviewer for their thorough reading of the manuscript and their positive words. We have now carefully responded to all concerns raised. We note especially that we have performed additional experiments to address the Reviewer’s primary concerns about our interpretation of the effects of FtsZ treadmilling perturbations on HT-PBP2B motion. These experiments largely support a model where FtsZ treadmilling does not directly regulate HT-PBP2B motion – instead the observed reduction in HT-PBP2B speed upon FtsZ treadmilling inhibition is likely caused by indirect effects. **We have added the new data (Supplementary Figure 17) and revised our discussion of this point (elaborated below in individual comments).**

1. Line one and throughout. The terminology “track” has been introduced in other papers in the field. However, the terminology becomes confusing in the model in Figure 5, where the “single-track” is the septal ring that consists of multiple separate tracks created by individual FtsZ filaments (Fig. 5a). The authors should consider labeling these multiple separate tracks as “sub-tracks” or “lanes” associate with separate FtsZ filaments to distinguish them from the idea of the larger “single-track” nomenclature.

As part of our responses to Comments 12 and 13 below, we have remade Figure 5 and revised our description of the model, removing the alternate uses of the term ‘track’. We have gone through the manuscript and ensured that the ‘track’ term is used in a way consistent with the wider field.

2. Line 25, 58, and throughout. As discussed below in point 12, the direct regulation of PG synthesis activity by FtsZ treadmilling is not firmly established in this paper and should not be stated as such a strong conclusion.

As part of our responses to Comments 12 and 13 below, we have now performed experiments to test whether FtsZ has a direct regulatory effect on sPG synthesis as our previous model had proposed. Our new results instead support a model where perturbations to FtsZ treadmilling reduce synthase speeds through an indirect effect. Our conclusion remains that there is some effect of FtsZ treadmilling on HT- PBP2B processive motion (Figure 3e), which we describe generically as ‘partial dependence’, although this likely results from indirect effects rather than direct regulation. **We have removed any mention of FtsZ treadmilling ‘regulating’ sPG synthesis throughout the manuscript.**

3. Line 72 and figures in general. After “induction level” please list the optimal IPTG concentration used. In general, there are other places where information can be included in figures and text so that readers do not need to flip around in the paper. For example, please go through and include the simplified genotypes for the strains in the figures, rather than just the strain numbers. As written, the reader has to flip to the strain table or other places in the manuscript to get this information, which can easily be included in the figures.

We have now specified the induction levels for HT-PBP2B and GFP-FtsZ in the first paragraph of Results and in the legend for Figure 1. Additionally, we have specified the JFX554 HaloTag ligand concentration used in the second paragraph of Results and the legend for Figure 1.

We have also gone through all figures and ensured that a simplified genotype is shown in both the figures themselves and the accompanying legends rather than only strain numbers (Supplementary Figures 2, 3, 4, and 6 of the original submission; now Supplementary Figures 2, 4, 7, and 16). Strain numbers are also shown in these figure legends for reference.

4. Line 72 after “cell morphology”. Is the cellular amount of HT-PBP2B known relative to the normal WT level, since HT-PBP2B is the sole source in these strains? Are we dealing with HT-PBP2B at normal WT levels or is HT-PBP2B over- or underexpressed in these experiments? If different from WT, how would HT-PBP2B impact interpretations?

We have included a Western blot quantifying the levels of HT-PBP2B expression across [IPTG] as Supplementary Figure 3 and have included discussion of this in Lines 74-79, reproduced here:

We constructed a strain that expresses a previously-characterized HaloTag (HT) fusion of PBP2B as a functional sole copy at its native locus from an IPTG-inducible promoter² (Methods; Supplementary Table 1). Induction of protein expression with 100 μM IPTG gave near-native cell morphology (Supplementary Figure 2), with HT-PBP2B levels at $\sim 67\%$ of native PBP2B levels (Supplementary Figure 3). We chose 100 μM IPTG induction of HT-PBP2B for experiments as higher induction levels did not produce significant changes in cell morphology (Supplementary Figure 2).

We expressed HT-PBP2B at 100 μM IPTG induction rather than using higher levels as this level is sufficient to recapitulate near-native morphology and thus normal cell division – for which cell length is a sensitive marker – while also giving optimal imaging conditions by minimizing the amount of unbound PBP2B - which is otherwise observed as background signal.

As a control, we have now measured HT-PBP2B speeds in this strain with 10-fold higher [IPTG] (1 mM). As shown in Supplementary Figure 3, this concentration of inducer results in HT-PBP2B expression $\sim 80\%$ that of native levels. The mean speed of the processive population we measure under this condition is 11.0 nm/s (95% CI [9.1, 13.1]), similar to what we measured with 100 μM IPTG induction (13.4 nm/s; 95% CI [12.7, 14.1]).

We now include this control experiment as Supplementary Figure 12 and mention it briefly in the text (Lines 169-171), reproduced here:

This result does not depend on protein expression levels, as inducing HT-PBP2B with a 10-fold higher concentration of IPTG resulted in similar speeds for processive HT-PBP2B molecules (Supplementary Figure 12).

5. Line 73. Is the GFP-FtsZ active? Does it perturb WT FtsZ function at all? Please clarify.

The GFP-FtsZ fusion in our SH147 strain is likely not fully active. However, we were expressing it only at low levels (0.075% xylose (50 μM) induction) to use as a marker for the position of the division septum.

Cell length is a sensitive marker for cell division defects, and so a way to determine whether our expression of GFP-FtsZ perturbs cell division (likely by perturbing FtsZ function) is to measure cell lengths across induction levels. We have now measured cell lengths across [xylose]. Up to 1.5% xylose (1 mM), we observe a very slight increase in cell lengths ($\sim 8\text{-}9\%$), indicating that even saturating expression of this GFP-FtsZ fusion in cells has a minor effect

on cell division. This is in contrast to the point mutant FtsZ^{G106S}, which shows dramatic increases in cell length (~62%) and Z-ring morphology, along with a substantial reduction in treadmilling speed (now shown in Supplementary Figure 16). This suggests that the level of GFP-FtsZ fusion we express throughout our experiments does not perturb native FtsZ function.

We have now provided this data on cell lengths across [xylose] in Supplementary Figure 2.

Additionally, as part of our response to Comment 10c we have now measured the treadmilling speed of FtsZ in the strain used in most experiments in the paper (HT-PBP2B, GFP-FtsZ, Δ hag) in rich media at 37°C using the same induction level (0.075% xylose). The median speed of FtsZ treadmilling we measure is 46 nm/s, which is comparable to our previous measurements in a different strain across multiple conditions.

We have now provided this new data on FtsZ treadmilling in our HT-PBP2B, GFP-FtsZ, Δ hag strain at our usual level of GFP-FtsZ induction (0.075% xylose) compared to our previous measurements from a strain expressing mNeonGreen-FtsZ as Supplementary Figure 5.

6. Fig. 1. The green GFP-FtsZ ring is used to find the plane for tracking. How thick is this ring relative to the resolution of this method? That is, if the plane of focus is moved up or down are new slices of tracks revealed. Please comment.

The FtsZ ring in vegetatively-growing *B. subtilis* cells is estimated by cryo-electron tomography to be

~50 nm thick (Khanna et al. *eLife* 2021). However, the depth of field of our imaging system (i.e. the axial distance over which any objects will be in focus) is ~300 nm. This means that when we are focused exactly on the middle of the septal ring in our vertical cell system, single-molecule spots will be in focus if they are within ± 150 nm from the septum. So, with this imaging method we will not be able to resolve multiple 'lanes' within the septal ring as depicted schematically in Figure 5 of our original submission.

When imaging, we choose a focal plane to maximize the number of division septa that are in focus, but there are always cells in the field of view where we are instead focused on a random part of the sidewall. This is the primary reason we used GFP-FtsZ as a septal marker – so we can filter out cases where we are not imaging HT-PBP2B molecules at division septa. We have observed that HT-PBP2B tracks that are away from the septal plane are nearly all diffusive (see response to comment 7f).

7. Fig. 1. A couple points about this figure. In general, there are additional interesting data and information here that are not brought out.

- a. For general readers, the authors should indicate that diagonal tracks in Fig. 1c indicate processive motion, whereas horizontal tracks indicate lack of motion (Fig. 1d). The authors should also indicate that when a diagonal track changes direction as in Fig. 1c, it indicates change in direction of the HT-PBP2B molecules, including reversals.*

We have now drawn lines over one diagonal track segment and one horizontal track segment in Figure 1c and listed the measured speeds, using the same colors as the fits in panel f that correspond to 'processive' and 'immobile'. We have also indicated direction changes in our kymographs with blue arrows. These are described in the legend accompanying the figure.

- b. It seems that most of the kymographs have two tracks, one diagonal (moving) and the other horizontal (non-moving), like Fig. 1c. Were kymographs detected that*

support the multiple sub-track arrangement shown in the model in Figure 5. If so, please include and highlight these.

We have observed a number of cases where processively-moving single-molecule tracks of HT-PBP2B appear to cross over one another. This suggests that there are divisive synthesis complexes that move out of plane from one another, as they pass by without apparently colliding and stopping. Since we used a very low labelling density of the JFX554 HaloTag ligand dye to observe single-molecule tracks, encountering two processive molecules that pass by one another like this should be somewhat rare. Since we see these events even with sparse labelling, we expect that this situation is relatively common in the unlabelled population.

We have now provided several examples of these ‘track crossing’ events as Supplementary Figure 8 and included mention of them in the text (Lines 113-116), reproduced here:

We also observed numerous cases of processive tracks apparently crossing one another (Supplementary Figure 8), suggesting that multiple divisive synthesis complexes exist in different lanes at the septal leading edge, slightly out of plane from one another.

- c. *What is the percentage of molecules that move processively without changing direction? How often do non-motile molecules start moving processively or processively moving molecule stop? Do processive or stopped molecules start moving diffusively?*

Under our standard imaging conditions (rich media, 30°C), we find that of processive track segments (N=570 total):

- 54 ± 4% (N=307) were terminal (i.e. ended in either photobleaching, dissociation, or end of acquisition period)
- 29 ± 3% (N=164) ended by a change in direction
- 12 ± 2% (N=68) ended in immobilization
- 5 ± 1% (N=28) ended in the molecule becoming diffusive
- 0.5 ± 0.3% (N=3) ended by a significant change in speed, but not change in direction

We have now listed all percentages in all experimental conditions comprehensively in the new Supplementary Table 2 and described them in the text (Lines 109-113), reproduced here:

Processively moving molecules showed a variety of noteworthy behaviours. Individual processive runs were usually terminal (54 ± 4%; mean ± SD; Supplementary Table 2), indicating that they ended in dissociation, photobleaching, or the end of the acquisition period. However, they often ended with a change in direction (29 ± 3%), or sometimes by becoming immobile (12 ± 2%) or changing to the fast-motion state (5 ± 1%; Supplementary Figure 6).

Regarding the second question: The Reviewer is asking if we can calculate the transition rates

$k_{immobile \rightarrow processive}$ and $k_{processive \rightarrow immobile}$, among others. The kinetics in this case are somewhat complex, as there are multiple competing pathways to leave a given state (e.g. a processive molecule can change direction, immobilize, photobleach, etc.). However, we can calculate these using the probabilities stated above as well as the lifetime of the state. For example, we can obtain the rate of a molecule in the processive state becoming immobile using

$$k_{processive \rightarrow immobile} =$$

$p_{processive \rightarrow immobile} / \langle t_{processive} \rangle$. With the probabilities above and a measured mean lifetime of 29 s, this yields:

- $k_{processive \rightarrow terminal} = 0.019 \pm 0.002 \text{ s}^{-1}$
- $k_{processive \rightarrow processive \text{ (direction change)}} = 0.010 \pm 0.001 \text{ s}^{-1}$
- $k_{processive \rightarrow immobile} = 0.0042 \pm 0.0006 \text{ s}^{-1}$
- $k_{processive \rightarrow diffusive} = 0.0017 \pm 0.0003 \text{ s}^{-1}$
- $k_{processive \rightarrow processive \text{ (speed change)}} = 0.0002 \pm 0.0001 \text{ s}^{-1}$

We have now listed all transition rates in all experimental conditions comprehensively in the new Supplementary Table 2. We have also demonstrated how we calculated these rates in the new Supplementary Note 1. Addition to the text is reproduced here (Lines 95-97):

The probabilities of transitioning between states and their associated rates for all conditions in this study are listed in Supplementary Table 2.

Regarding the third question: We have observed a number of cases where PBP2B molecules transition states from processive or immobile to diffusive, and vice versa. This further highlights the kinetic

complexity of this system. **We have now provided examples of these ‘state change’ events as Supplementary Figure 6 and made reference to them in the text (Lines 94-95), reproduced here:**

Individual HT-PBP2B molecules were also capable of transitioning between states (Supplementary Figure 6).

d. What is the relative distribution of septal ring HT-PBP2b molecules that are moving processively, not moving, or diffusively moving. This interesting information is buried in the frequencies in Fig. 1f-h, but is never pulled together.

The relative populations of immobile, processive, and diffusive molecules cannot be directly obtained from Figure 1f-h, as the counts recorded in those histograms are speeds, which are independent of lifetimes. That is, if we recorded many speeds of some state that was very short-lived (as was the case for the diffusive population), the histogram of speeds would make it appear that this state makes up a greater fraction of the population at any given point in time than it really does. However, as we can also measure the lifetimes of these segments, we can use this additional information to calculate these fractions.

For our standard conditions, we calculate that, for a given point in time, HT-PBP2B can be expected to be found:

- $38.1 \pm 0.4\%$ in the immobile state
- $59.0 \pm 0.6\%$ in the processive state
- $3.0 \pm 0.1\%$ in the diffusive state

We now list all of the fractions of states for each experimental condition comprehensively in the new Supplementary Table 3 and describe them in the text, reproduced below (Lines 97-99):

Based on the measured lifetimes of these states, an HT-PBP2B molecule under our standard conditions exists in the immobile state $38.1 \pm 0.4\%$ of the time, the processive state $59.0 \pm 0.6\%$, and the fast-moving state $3.0 \pm 0.1\%$ (Supplementary Table 3).

e. It seem that the calculated lifetime of tracks include motion in more than one direction. Please specify this.

The lifetime reported (Line 102) is the lifetime of the immobile state only (speed < 4 nm/s).

That is, if a HT-PBP2B molecule was moving processively and then stopped, we only measured this lifetime from the time it stopped until the time it either began moving again, or the signal was lost (due to either photobleaching or dissociation). The goal here was to compare the lifetime of this immobile state to that of (also immobile) FtsZ monomers reported previously.

We have reworded this in the main text (Lines 100-104) to make it clearer that this is the lifetime of the immobile state only, and not of full tracks with multiple states. The change is reproduced here:

We considered that the immobile population could represent molecules bound to the middle of treadmilling FtsZ filaments, as suggested previously for FtsW in *E. coli*¹¹. However, the average lifetime of the immobile state of HT-PBP2B molecules (48 ± 3 s (mean \pm SEM); Supplementary Figure 7) is substantially longer than the reported lifetime of FtsZ monomers at the division septum in *B. subtilis* (8.1 s)⁷, making this situation unlikely.

f. *Are diffusing HT-PBP2B molecules detected outside of the septal ring area? If so, do these molecules show the higher velocity characteristic of free diffusion? Please comment.*

As discussed above in our response to comment 6 about the depth of field in our system, in any given field of view we will be focused on regions of the cell sidewall outside the septum.

We have now gone through our data, selecting for those cells where we appear to be focused on the cell sidewall rather than the septum (based on the GFP-FtsZ signal) and repeated our analysis of linear speeds. The speeds in this case show the diffusive population almost exclusively. The mean speed of this population (mean of all speeds >60 nm/s) is 230 nm/s, which is comparable to the diffusive population at the septum (180 nm/s).

We have now provided a Supplementary Figure 11 showing the speeds of HT-PBP2B molecules outside the septal area and description in the text (Lines 128-133), reproduced here:

We next wondered what the broad fast-speed population was. This population resulted from short back-and-forth tracks that appeared to show diffusive rather than processive motion (Supplementary Figure 10a). These tracks were often located at larger radii than immobile or processive tracks (Supplementary Figure 10b) and were almost the exclusive motion pattern outside the septal ring area (Supplementary Figure 11), suggesting that they are not typically present at the septal leading edge and therefore unlikely to be involved in sPG synthesis.

We noted in our manuscript that the diffusion coefficient we measured for PBP2B is similar to that of

E. coli PBP2 as measured by Özbaykal et al. 2020 (Line 141), although we neglected to mention one point about this that is relevant here. The diffusion coefficient we measured is substantially lower than would be expected for a protein with a single transmembrane helix such as PBP2B. One explanation is that during cell division PBP2B remains part of the much larger divisome core complex even when unbound from the septal leading edge. This would necessarily have a lower diffusion coefficient.

We have now added explanations for the low diffusion coefficient to the text (Lines 140-144), reproduced here:

The low diffusion coefficient is similar to that previously measured for the *E. coli* elongasome transpeptidase PBP2¹⁸, and suggests that diffusive HT-PBP2B molecules may principally exist as part of large multi-protein complexes (i.e. the divisome core complex). Alternatively, diffusive HT-PBP2B molecules may experience substantial molecular friction through transient interactions with the cell wall.

8. Line 91. What is the basis for the assumption that the non-moving molecules are bound to the cell wall? How do you envision this binding? Incomplete TP half-reactions? As alternatives, the non-motile molecules may simply not be catalytically active or may be trapped in incomplete or aberrantly assembled complexes. Please qualify this speculation more.

The basis for our assumption was that the divisome synthesis complex is known to bind the cell wall, and this is one of the few structures static enough that we expect could keep a molecule immobile over the 48 s lifetime that we measured. We made this claim at this phenomenological level rather than at the level of the mechanism of binding.

However, there are several possibilities, including those that the Reviewer has enumerated. One attractive possibility was proposed recently by Özbaykal et al. regarding how the elongasome transpeptidase PBP2 may bind directly to the cell wall in *E. coli* (Özbaykal et al. *eLife* 2020). These authors noted that PBP crosslinking requires bringing together donor peptides on nascent glycan

strands and acceptor peptides in the cell wall. So, in our case an immobile PBP2B molecule, possibly together with the rest of the divisome synthesis complex, could be bound to an acceptor peptide in the cell wall, prior to initiation of sPG synthesis.

We have included further discussion in Lines 105-108, reproduced here:

It is unclear how such binding would occur, although one attractive possibility is that immobile PBP2B molecules are bound to acceptor peptides in the cell wall awaiting the emergence of nascent glycan strands to crosslink, as proposed recently for the *E. coli* elongasome transpeptidase PBP2¹⁸.

9. In Fig. 1, Is the drop in speed of the sub-diffusing molecules in the presence of antibiotics significant, and if so, why does it happen?

We have not observed a significant drop in speed for the sub-diffusive (>100 nm/s) population with the addition of penicillin G or fosfomycin. If we take all speeds >60 nm/s to ensure we include the full 'sub-diffusive' population, these are:

- Untreated: speed = 181 ± 82 nm/s (mean \pm SD)
- Penicillin G: speed = 188 ± 103 nm/s
- Fosfomycin: speed = 194 ± 101 nm/s

However, Figure 1f-h does show that there may be a slight decrease in the frequency of the diffusive population in the presence of these antibiotics. However, as stated in the response to point 7d above, the size of this population is not well represented by these histograms, since they show the speeds rather than lifetimes. Accounting for the lifetimes of the states, we see that the diffusive population makes up:

- $3 \pm 2\%$ of the untreated population
- $2 \pm 2\%$ of the penicillin G population
- $2 \pm 3\%$ of the fosfomycin population

So, the population of the diffusive state is not substantially altered by the presence of these antibiotics.

The fraction of each state for each experimental condition is now represented by Supplementary Table 3.

10. Figure 2 raises a variety of important questions that need to be addressed or commented on.

30- a. *Why does the FtsZ treadmilling speed not show a temperature dependence? The average treadmilling speed should be temperature dependent, because it depends on an enzymatic reaction. Please explain this.*

The Reviewer raises an interesting point here. The values presented here showing the temperature independence of FtsZ treadmilling speed come from our previous paper (Whitley et al. *Nature Communications* 2021), yet this is indeed surprising. We can estimate the rates of polymerization and depolymerization using the Eyring equation:

$$k = \frac{k_B}{h} \frac{\Delta S^\ddagger}{H^\ddagger + k_B T}$$

If we are comparing the rates across two temperatures, we can rearrange this expression to:

$$\frac{k_1}{k_2} = \frac{1}{e} e^{\frac{\Delta H^\ddagger}{RT_1} - \frac{\Delta H^\ddagger}{RT_2}}$$

along with estimates for ΔH^\ddagger measured for *E. coli* FtsZ GTPase / depolymerization *in vitro* (Concha- Marambio et al. *PLoS ONE* 2017). Using the published value of $\Delta H^\ddagger = 98.4$ kJ/mol for depolymerization, we find that an increase from 30°C to 37°C should cause an increase of ~2-fold in this rate, and hence likely in treadmilling speed. However, we should note that these numbers should be taken cautiously, as this enthalpy value, along with the reported activation entropy in that publication (134 J/K/mol), predict a depolymerization rate of 480 s⁻¹ at 30°C. If this corresponds to the rate of treadmilling, then with an approximate FtsZ diameter of ~2 nm, this predicts a treadmilling speed of 960 nm/s, which is far from what we observe *in vivo* (~40 nm/s). Nevertheless, even with an activation enthalpy ΔH^\ddagger half the reported amount, we would still expect an increase of ~60% in treadmilling speed from 30°C to 37°C.

To test how far this temperature independence extrapolates, we have now measured FtsZ treadmilling speed at 21°C. We do observe a 32% reduction in treadmilling speed from 30°C, however this is substantially less of a reduction than would be expected from the Eyring equation (68%).

These results compared with the Eyring equation predictions suggests that bacterial cells actively regulate the polymerization dynamics of FtsZ filaments, as the temperature independence of treadmilling speed is not explained by chemical physics alone.

We have included our new measurements of FtsZ treadmilling speed at 21°C as Supplementary Figure 13 and have included discussion of this in the text (Lines 219-229; reproduced below) with an accompanying Note (Supplementary Note 2) about our predictions.

It is surprising that FtsZ treadmilling speed is independent of temperature, as it depends on an enzymatic reaction. Using the Eyring equation with activation enthalpies

measured previously for *E. coli* FtsZ *in vitro*²⁰, we predict that an increase of 30°C to 37°C should result in a ~2-fold increase in treadmilling speed (Supplementary Note 2), which we have not observed. To test how far this temperature independence extrapolates, we measured FtsZ treadmilling speed at 21°C using a previously characterized strain (bWM4; Supplementary Table 1) expressing mNeonGreen-FtsZ from an IPTG-inducible promoter (Supplementary Figure 13), grown in rich media. Treadmilling speed was 32% slower than it was at 30°C, although this is a substantially smaller decrease than the 68% predicted from the Eyring equation. This suggests that bacterial cells may actively regulate the polymerization dynamics of FtsZ filaments, as the lack of temperature dependence of treadmilling speed is not explained by chemical physics alone.

b. Why are the distributions so different in rich medium at 37 degrees? Remarkably, the HT-PBP2B distribution pancakes out. Why is this? Why is the frequency of FtsZ treadmilling apparently much lower at 37 degrees in rich medium compared to the other three panels? Please offer some possible explanations for this singular behavior, which the text argues may have affected previous interpretations of data.

First, regarding the Reviewer's question about the frequency of FtsZ treadmilling (amplitude of the distribution): As stated in the legend for Figure 2, the amplitudes of the FtsZ treadmilling distributions were scaled to match the heights of the HT-PBP2B populations for better comparison. We decided to represent it this way so that comparisons of the means and variances of the FtsZ and HT-PBP2B distributions can be more easily done by eye. So, the reason the frequency (i.e. amplitude) of the FtsZ treadmilling distribution is low in the "rich medium 37°C" panel is that it was scaled to match the low amplitude of the processive HT-PBP2B population. To make this clearer, we have added a second y-axis for FtsZ treadmilling speed on the right-hand sides of the panels.

Second, regarding the Reviewer's question about the 'pancaking' of the HT-PBP2B distribution under this growth condition: This is a keen observation that we have also wondered about. The large variance of this population under these conditions is reproducible: We observe a similarly large spread for each individual replicate we performed.

We had initially suspected that this increase in variance was the result of a Poisson process – which is quite common in stochastic single molecule dynamics, where the variance of a distribution increases linearly with its mean. However, the variances of the processive HT-PBP2B populations under the four conditions shown in Figure 2 increases faster than a Poisson process, scaling approximately exponentially with the mean. We do not yet know what mechanism underlies this increase in variance. In the future we aim to pursue computational modelling of divisome synthase dynamics which might provide insight into this point. **We have included some discussion of the increased variance in the text (Lines 188-198), reproduced here:**

The variance of the processive HT-PBP2B population under the fastest growth condition is substantially higher than that of other conditions (Figure 2a). The reason for this increase is unclear, although it was reproducible: each of the four biological replicates we performed under this condition had a similarly large variance. This increase is larger than the dependence predicted by a Poisson process, which is commonly observed in single molecule dynamics. Furthermore it does not seem to result from perturbed FtsZ treadmilling in the particular strain used here, as the speed distribution for FtsZ treadmilling under these conditions was comparable to those of previous measurements (Supplementary Figure 5). It is possible that the increased variance in speeds under this condition reflects increased variation in the production or local availability of the lipid II substrate used by the divisome synthesis complex. However, this is currently difficult to investigate experimentally.

- c. The FtsZ treadmilling is taken from previously reported experiments, presumably in different strains than the one expressing HT-PBP2b. If it is a different strain, was the treadmilling rate spot checked and could it be relevant to point b above? Please comment.*

We have now spot-checked the treadmilling speed of FtsZ in our HT-PBP2B, GFP-FtsZ, Δ hag strain under the exact conditions in question (rich media 37°C, 100 μ M IPTG, 0.075% xylose). The speed distribution we obtain is comparable with what we have measured previously for a

strain expressing mNeonGreen- FtsZ. The speeds in our strain here are slightly higher (46 nm/s vs. 40 nm/s) and have slightly increased variance ($387 \text{ nm}^2/\text{s}^2$ vs. $322 \text{ nm}^2/\text{s}^2$), but this does not appear to account for the increase in variance we have measured for HT-PBP2B speeds ($39 \text{ nm}^2/\text{s}^2$ in rich media 30C vs. $205 \text{ nm}^2/\text{s}^2$ in rich media 37C).

We now present our results for FtsZ treadmilling in this strain under the exact conditions in question as Supplementary Figure 5 and reference it in the text (Lines 192-195), reproduced here:

Furthermore, it does not seem to result from perturbed FtsZ treadmilling in the particular strain used here, as the speed distribution for FtsZ treadmilling under these conditions was comparable to those of previous measurements (Supplementary Figure 5).

11. Fig. S5 and the correlations in Fig. 2a and 2c are particularly interesting and nice; also, the results showing multimeric complexes later.

We thank the reviewer for their encouraging words.

12. Lines 174-198. *There are several issues with the interpretation of the data presented in Figure 3. The data show that there is a moderate (about 40%) drop in HT-PBP2B processive speed when FtsZ treadmilling is inhibited partially (but enough to cause large morphology differences (Fig. S6)) or completely (PC19 compound). These data are convincing and correct the previous result that indicated no movement in the presence of the PC19 compound.*

However, the conclusion that the slower HT-PBP2B velocity is a direct effect (“depends on”) FtsZ treadmilling is not justified, because the change could be an indirect effect.

Our conclusion was that HT-PBP2B speeds ‘partially depend on’ FtsZ treadmilling, where we used the term ‘partial dependence’ in the generic sense that HT-PBP2B speeds are not fully independent of FtsZ treadmilling. The latter scenario would have resulted in no change to HT-PBP2B speeds when FtsZ treadmilling is abolished, which is not what we have observed (Figure 3e). However, we fully agree with the Reviewer that there are alternative explanations for this partial dependence, including that it results from an indirect effect rather than the more direct effect we depicted in the ‘broken track’ model (Figure 5 of our first submission).

In response to points a-c below, as well as Comment 13, we have now performed further experiments to test whether the reduction in synthase speeds we observe is better explained by the model we proposed initially or by an indirect effect as the Reviewer suggests. In short, we imaged the motion of HT-PBP2B molecules along septa in horizontally-oriented cells and investigated the degree of off-axis motion in unperturbed cells compared to those treated with PC190723 or those expressing the FtsZ^{G106S} mutant. Although we do measure a slight but significant increase in off-axis localizations in the PC190723 and FtsZ^{G106S} cases, the size of the effect is only ~2 nm, which is unlikely to be biologically meaningful. This suggests to us that the partial dependence of synthase speeds on FtsZ treadmilling is more likely an indirect effect as the Reviewer suggests.

A description of the changes made to the manuscript from these new experiments can be found in our response to Reviewer 1 Comment 5.

- a. *In particular, the paper nicely shows that HT-PBP2B velocity depends on the rate of PG synthesis, which is dependent on cell metabolism (e.g., fosfomycin (Fig. 1e); and growth rate (Fig. 2b)). Limiting or eliminating FtsZ treadmilling is bound to be highly pleiotropic and to set off all kinds of stress responses, including ones that may limit PG precursor synthesis, which will reduce HT-PBP2B velocity.*

The Reviewer brings up an excellent point here. It is unclear what other processes in the cell will be affected by the perturbations to FtsZ treadmilling that we have done. Conceivably, these perturbations could trigger stress responses in the cells that would affect metabolism. **We have now mentioned this in the discussion (Lines 353-361), reproduced here:**

The underlying mechanism for the reduction in synthase complex speeds upon FtsZ treadmilling perturbation remains unclear. It is possible that the disruption to FtsZ treadmilling leads to an alteration of transient interactions between FtsAZ filaments and the divisome synthesis complex, leading to increased molecular friction. However, we consider this unlikely, as both a reduction and total arrest of FtsZ treadmilling produced similar effects on HT-PBP2B speeds (Figure 3e). The reduction in HT-PBP2B speeds may instead be an indirect effect of disrupting FtsZ treadmilling. We have shown that the speeds of the divisome synthesis complex depends on cell metabolism (Figure 2b). It is plausible that severe perturbations to an essential and abundant protein such as FtsZ could affect metabolism (e.g. through a stress response) and hence indirectly reduce synthase speeds.

- b. *In addition, if HT-PBP2B speed is regulated by FtsZ treadmilling, then why does partial limitation of FtsZ treadmilling show the same dependence as complete inhibition of FtsZ treadmilling? It seems that the proposed regulation would be graded.*

The Reviewer raises a good point here, which we have also considered. The lack of such a linear correlation between HT-PBP2B speed and FtsZ treadmilling speed is part of the reason we had proposed the ‘broken track’ model shown in Figure 5 of our original submission. By this model, any substantial perturbation to FtsZ treadmilling will disrupt order at the division ring and lead to a similar reduction in PBP2B speeds. However, as discussed above we now consider indirect effects to be the more likely explanation and have revised the model.

- c. *Last, why does the distribution of processively moving HT-PBP2B shift to lower speeds when FtsZ is partially inhibited in the mutant compared to completely inhibited by the PC19 compound?*

We were initially surprised as well to find that the mean speed of HT-PBP2B for the partial inhibition (7 nm/s; FtsZ^{G106S}) was slightly lower than that for total inhibition (8 nm/s; PC190723). However, as can be seen in Figure 3e, the two values are within error of one another. If we consider only the population in the ‘processive’ range (2 < speed < 40 nm/s), the distributions are not significantly different ($p = 0.56$ using paired t test). This is in contrast to the differences between the untreated WT processive population and PC190723-treated processive population ($p = 1.9 \times 10^{-7}$) or between the untreated WT processive population and FtsZ^{G106S} processive population ($p = 3.9 \times 10^{-5}$).

The idea that treadmilling somehow is regulating HT-PBP2B speed directly as stated in the Abstract, here, the Discussion, and Figure 5 is not strongly supported by this single set of data in stressed cells. These sections should be rephrased and revised to include alternative interpretations.

As also stated in our response to Comment 2, we agree with the reviewer that ‘regulate’ is not the best description of the interaction of FtsZ treadmilling and HT-PBP2B processive motion. Furthermore, in light of additional experiments (described above) to test the broken road model for regulation of HT- PBP2B dynamics by FtsZ, we conclude that regulation of HT-PBP2B by FtsZ dynamics via direct molecular interactions with FtsAZ is unlikely. **We have rephrased the sections in question and included alternative interpretations that the**

partial dependence of HT-PBP2B speed on FtsZ treadmilling we observe may result from indirect effects (Lines 353-361; reproduced above in reponse to Comment 12a).

13. Line 236. Was there a marked increase in non-linear HT-PBP2B movement detected in the FtsZ- G106S or PC19 experiments? The tracks shown in Fig. 3 do not seem to be different from those in Fig. 1c and elsewhere. In mutants of FtsZ filament bundling proteins, is HT-PBPB velocity slower? This seems like a corollary of this model that could be tested; although again, more severe bundling mutants may be pleiotropic on metabolism and PG synthesis.

As stated in our response to Comment 12 above, we have now investigated non-linear (off-axis) motion of processive HT-PBP2B molecules along the Z ring and have not observed a meaningful increase with

PC190723 or FtsZ^{G106S}. We have updated the text and figures as described in our response to that comment.

14. Line 263. Has the TIRFm experiment been repeated with these strains and conditions? Wasn't the motion determined previously in cells at different stages of septation and not just in cells at late stages of septation? For a population that contains lots of early divisional cells, why would the previously TIRFm measurement be biased and show no motion at all? This does not seem like a strong argument. Please consider revising or omitting.

The single-molecule tracking of HT-PBP2B that was observed in the previous publication (Bisson-Filho et al. *Science* 2017) was done with TIRF illumination only, which means that only motion at the bottom slice of the cell (~100 nm) will be seen. This will restrict observation of HT-PBP2B motion to only early divisional cells (in mid- to late-constriction, the septal leading edge is too far from the coverslip surface to be imaged with TIRF). This was the rationale behind our explanation.

However, the Reviewer raises a valid point in that our explanation predicts that HT-PBP2B molecules will primarily (or exclusively) be immobile (speed < 4 nm/s) at large septal diameters (i.e. early divisional cells). We have now done further analysis and do not observe a dependence of HT-PBP2B speeds on septal diameter.

We have now provided a plot of HT-PBP2B speed vs. septal diameter as Supplementary Figure 9 and referenced it in the text (Lines 116-118), reproduced here:

The speeds of processive molecules were independent of septal diameter (Supplementary Figure 9), suggesting that their dynamics remain consistent throughout active constriction.

Additionally, as part of our response to Comments 12 and 13 above, we have now repeated the TIRF imaging of HT-PBP2B in our two-colour strain post-PC19 treatment (Supplementary Figure 17 and Supplementary Video 15). We clearly observe processively-moving HT-PBP2B molecules at dividing septa under these conditions. It is unclear to us why our collaborators did not observe this in the previous study. We speculate that these processively moving PBP2B population may have been missed due to small sample size. **We have revised our previous explanation for our study's disagreement with prior results (Lines 334-341), reproduced here:**

Previously, our collaborators observed that arrest of FtsZ treadmilling by the antibiotic PC190723 abolished processive HT-PBP2B motion², but here we find that the effect of arresting FtsZ treadmilling is to slow HT-PBP2B processive motion rather than stop it. One difference between these studies is that cells were imaged here with VerCINI while the previous study imaged horizontally- oriented cells with Total Internal Reflection Fluorescence (TIRF) illumination. However, as part of this study we also imaged horizontally-oriented cells with TIRF and observed processive motion at division septa post-PC190723 treatment (Supplementary Video 15). It is possible that processive motion was not observed in the previous study due to small sample size.

15. Line 275. The examples presented indicate major differences and not “minor” differences as stated. Please change to: “there may remain differences”. Also, please see point 12 above, supporting further qualification of the model presented throughout.

We have removed the word “minor” as suggested.

16. Line 287. Please add: HT-PBP2B in the septal ring (please see point 7f, above).

We have added this where requested.

17. Line 423. Remove: “in some cases”, since all of the diagrams seem to use this format.

We have removed these words.

18. Fig. S5. What are the times on the kymographs?

We have added the times to these kymographs.

19. Fig. S9. A lot of these tracks look horizontal indicating non-motile. It would be helpful to indicate in all of the kymographs which tracks are considered to represent processively moving or non-motile molecules.

We note that the focus of Supplementary Figure 9 of the original submission (now Supplementary Figure 20) is the tracks which show intensity steps characteristic of molecule binding/ unbinding. **We have now colour coded those tracks cyan for motile and magenta for non-motile. We have also done this for the other Supplementary Figures showing stoichiometry results.** To avoid clutter we have not annotated any of the tracks that do not show binding/ unbinding dynamics.

Reviewer #3 (Remarks to the Author):

The manuscript by Whitley et al. analyzes the dynamics of peptidoglycan synthesis during septation in Bacillus subtilis. By measuring PBP2B dynamics, the authors report that PBP2 moves at speed that is different than FtsZ treadmilling speed. The authors propose that the motion of PBP2B is mediated by peptidoglycan synthesis, and not by FtsZ treadmilling. Since they observed that the speed of PBP2B was affected when FtsZ treadmilling was inhibited, they conclude that PBP2B speed is only partially dependent on FtsZ treadmilling. The study proposes an evolved model for regulation of peptidoglycan synthesis during cell division in Gram-positive bacteria in which cell wall synthesis proceeds via a single population of a multimeric complex. The study certainly addresses an important question that should be of

42broad interest to bacterial cell biologists. However, I have a several experimental suggestions for additional controls that may strengthen the authors conclusions.

We thank the Reviewer for their positive comments and suggestions of further experimental controls. We have now addressed each point raised. In particular, we now provide experimental results on the stoichiometry of FtsW to strengthen our conclusion that the divisome synthesis complex is multimeric.

Major comments:

- 1. Figure S3. The authors characterize strain SH147 (inducible FtsZ-sGFP and HALO-pbp2B) to show that the growth curves are largely unaffected by induction of both constructs. Please include micrographs and analysis of cell lengths to see if cell morphology is affected by induction of these constructs. The subsequent data in the paper may still be valid even if morphology is affected, but*

the reader should be aware of the physiological consequences for the cell of producing these molecules.

We have already provided an analysis of cell lengths across [IPTG] to show the effect of HT-PBP2B expression levels on cell morphology in Supplementary Figure 2. However, in Supplementary Figure 2 of our first submission, the cell lengths shown were obtained from cells grown in 16 mm diameter culture tubes, unlike the rest of the measurements presented in the paper where cells were grown in 125 mL conical flasks. This was the result of a communication error. We have observed separately that growing cultures in tubes vs. flasks can have a substantial effect on cell morphology. Since all data presented in the other figures were obtained from cells grown in flasks, we have repeated the measurement of cell lengths across [IPTG] with cells grown in flasks.

We have now updated Supplementary Figure 2 with the new measurements of cell lengths across [IPTG]. As requested, we have also now provided the accompanying micrographs alongside this as Supplementary Figure 2a-b. Additionally, we now provide analysis of cell lengths across [xylose] to show the effect of GFP-FtsZ expression levels on cell morphology as Supplementary Figure 2d.

2. Fig. 2. The presented results for speeds of PBP2B and FtsZ are consistent with the authors' model of PG synthesis being largely independent on FtsZ treadmilling. That said, since FtsZ treadmilling depends on GTPase activity, and I would assume that GTPase activity of FtsZ is dependent on temperature, do the authors have a sense of why FtsZ treadmilling speed does not appreciably vary across the different conditions tested? Do the authors have a different (perhaps genetic) method of varying FtsZ treadmilling speed to demonstrate that PBP2B speed is unaffected under that condition?

We have already provided results showing the effect a genetic mutant of FtsZ that has reduced treadmilling (FtsZ^{G106S}) speed on HT-PBP2B dynamics. As shown in Figure 3, HT-PBP2B speed is affected by the impairment of FtsZ treadmilling by both this mutant and by the inhibitor PC190723. In response to Comment 4 below, we have now also measured the treadmilling speed of FtsZ^{G106S} in our strain that expresses HT-PBP2B and confirm that it is slower than WT.

Regarding temperature, the Reviewer raises an interesting point. Please see our response to Reviewer 2's Comment 10a above on the same subject.

3. A major question that is not addressed by the manuscript is “what drives PBP2B movement in the septum?”. One possibility is the enzymatic activity of the complex due to the availability of substrates or influence of temperature. Can the authors analyze the dynamics of PBP2B in the presence of antibiotics that target PBP2B activity or test the behavior of PBP2B variants that display reduced enzymatic activity?

The question of what actually drives PBP2B movement is a very interesting one. In this study we show clearly that divisome motion is associated with and likely driven by sPG synthesis. However the detailed molecular mechanism underlying divisome processive motion is beyond the scope of this manuscript. We note that this has not yet been resolved either for elongasome proteins in ~12 years of study or for divisome proteins for ~6 years. Several papers (e.g. Garner et al. *Science* 2011, Domínguez-Escobar et al. *Science* 2011, Yang et al. *Nature Microbiology* 2021) strongly support a model where that the energy for this processive motion comes from the PG synthesis reaction itself – either the transglycosylase activity of the SEDS proteins or the transpeptidase activity of the PBPs.

Resolving the detailed molecular mechanism by which SEDS-bPBP PG synthesis activity leads to processive motion will require substantial *in vitro* structural biology efforts and possibly molecular dynamics simulations.

Regarding the Reviewer's question about 'the dynamics of PBP2B behaviour in the presence of antibiotics that target PBP2B activity', we already show the results of such experiments in Figure 1c-

h. In Figure 1d and g, we show the dynamics of PBP2B in the presence of penicillin G, which directly binds the active site of PBP2B, while in Figure 1e and h, we show PBP2B dynamics in the presence of fosfomycin, which binds an enzyme upstream in the cell wall synthesis pathway (MurA) to deplete the substrate pool for PBP2B. This then shows the effects of two separate antibiotics that affect PBP2B activity with separate modes of action. These two experiments show that sPG synthesis is required for, and likely drives, divisome processive motion. This is further supported by the strong correlation between PBP2B speed and cell growth rate. These findings mirror previous studies on the homologous elongasome complex, where similar perturbations were used to establish that elongasome processive motion is driven by PG synthesis (Garner et al. *Science* 2011, Domínguez-Escobar et al. *Science* 2011). We therefore respectfully suggest that these multiple perturbation experiments specifically targeting different aspects of PBP2B and divisome enzymatic activity already substantially address the reviewer's concerns.

Regarding the second half of the Reviewer's suggestion, to 'test the behaviour of PBP2B variants that display reduced enzymatic activity', unfortunately there are no known mutants of *B. subtilis* PBP2B with reduced activity known, apart from a mutant which has catalytic activity fully ablated (Sassine et al. *Molecular Microbiology* 2017). However, in cells expressing this catalytically dead mutant, a partially redundant enzyme (PBP3) is able to take over as the divisome transpeptidase (Sassine et al. 2017), and so an experiment to look at this mutant would likely not produce clear results. It is worth noting that our experiments in Figures 1d-f are unaffected by the existence of this secondary enzyme, as both antibiotics used will also inhibit PBP3.

We have now included discussion in the text about the evidence that sPG synthesis drives PBP2B/ divisome processive motion, and briefly mention the possible energy sources powering divisome processive motion (Lines 362-368), reproduced below:

Our results, along with those in multiple other organisms^{5,23}, strongly support a model where the processive motion of sPG synthases is driven exclusively by sPG synthesis. Our observation that treatment with antibiotics targeting either synthase

activity or lipid II precursor synthesis prevents processive motion (Figure 1c-h) suggests that the sPG insertion reaction itself may provide the required energy. A similar mechanism was previously proposed for elongasome synthases^{24–26}. Further work is required to understand the molecular mechanism by which SEDS-bPBP PG synthesis activity leads to processive motion.

4. *Fig. 3e. The authors report processive PBP2B speed as a function of FtsZ treadmilling speed, but it is unclear if the authors independently measured the treadmilling speed of the G106S variant in this study or if they simply used a previously published result. Since FtsZ treadmilling is a central part of the author's argument, please formally measure the treadmilling speed of G106S, in this study if it was not done already.*

As stated in the legend for Figure 3 in our first submission, we had used a previously published result for this value (from Bisson-Filho et al. *Science* 2017). However, we agree that it is sensible to measure it in our own lab as part of this study.

We have now re-measured the treadmill speed of FtsZ^{G106S} under our specific conditions and present the results in Supplementary Figure 15. We have also now updated the mean and 95% confidence interval in Fig. 3e with this new measurement.

For clarity, we note that while several other FtsZ speed measurements (Figure 2) were results presented in a previous study (Whitley et al. *Nature Communications* 2021), all of those measurements were conducted in our lab under identical conditions to those of this study and therefore do not need to be repeated.

5. The authors' conclusion that the divisome synthesis complex is multimeric is based solely on the fluorescence intensity plots in Fig. 4a. If the machinery that synthesizes septal PG is indeed multimeric, would the authors be able to measure similar behavior with another member of the complex (for example, FtsW), or via a dual labeling experiment with PBP2B and FtsW?

The Reviewer raises a valid point here. To further test our conclusion that the divisome synthesis complex is multimeric, we have now repeated the VerCINI experiments with a previously published HaloTag-FtsW strain gifted to us by the Garner lab (Harvard). We also see fluorescence drops and jumps similar to that of our HaloTag-PBP2B data, strengthening our conclusion that the divisome synthesis complex as a whole is multimeric, and not solely PBP2B.

We now provide some example kymographs showing evidence for higher-order oligomers of HT- FtsW in Supplementary Figure 21, and describe in the text (Lines 294-300), reproduced here:

We wondered whether this behaviour was unique to PBP2B, or if it was a more general feature of divisome synthesis complex proteins. We repeated these measurements with a strain expressing a HaloTag fusion of the transglycosylase FtsW (HT-FtsW; Supplementary Table 1) that together with PBP2B forms the core of the synthesis complex. HT-FtsW displayed similar fluorescence drops and jumps as HT-PBP2B (Supplementary Figure 21), suggesting that the divisome synthesis complex is

multimeric. The effect of stoichiometry on divisive activity and dynamics will be followed up in future work.

Decision Letter, first revision:

Message: Our ref: NMICROBIOL-23061536B

13th February 2024

Dear Dr. Whitley,

Thank you for your patience as we've prepared the guidelines for final submission of your Nature Microbiology manuscript, "A one-track model for spatiotemporal coordination of *Bacillus subtilis* septal cell wall synthesis" (NMICROBIOL-23061536B). Please carefully follow the step-by-step instructions provided in the attached file, and add a response in each row of the table to indicate the changes that you have made. Ensuring that each point is addressed will help to ensure that your revised manuscript can be swiftly handed over to our production team.

We would like to start working on your revised paper, with all of the requested files and forms, as soon as possible. My apologies that it took some time to get this checklist to you, however we would really like to try to expedite the rest of the process and publish this study in the April issue of the journal alongside the other back-to-back submission. As a result it would be great if you could look into this as soon as possible and resubmit within a week. I really hope that this is possible - do let me know if you foresee any delays. In the meantime I will be in contact with our production team to make sure that your study is a priority going down the line.

In recognition of the time and expertise our reviewers provide to Nature Microbiology's editorial process, we would like to formally acknowledge their contribution to the external peer review of your manuscript entitled "A one-track model for spatiotemporal coordination of *Bacillus subtilis* septal cell wall synthesis". For those reviewers who give their assent, we will be publishing their names alongside the published article.

2Nature Microbiology offers a Transparent Peer Review option for new original research manuscripts submitted after December 1st, 2019. As part of this initiative, we encourage our authors to support increased transparency into the peer review process by agreeing to have the reviewer comments, author rebuttal letters, and editorial decision letters published as a Supplementary item. When you submit your final files please clearly state in your cover letter whether or not you would like to participate in this initiative. Please note that failure to state your preference will result in delays in accepting your manuscript for publication.

Cover suggestions

COVER ARTWORK: We welcome submissions of artwork for consideration for our cover. For more information, please see our guide for cover artwork.

Nature Microbiology has now transitioned to a unified Rights Collection system which will allow our Author Services team to quickly and easily collect the rights and permissions required to publish your work. Approximately 10 days after your paper is formally accepted, you will receive an email in providing you with a link to complete the grant of rights. If your paper is eligible for Open Access, our Author Services team will also be in touch regarding any additional information that may be required to arrange payment for your article.

Please note that *Nature Microbiology* is a Transformative Journal (TJ). Authors may publish their research with us through the traditional subscription access route or make their paper immediately open access through payment of an article-processing charge (APC). Authors will not be required to make a final decision about access to their article until it has been accepted. Find out more about Transformative Journals

Best regards,

Reviewer #1:

Remarks to the Author:

The authors have done an excellent job responding to the reviewers' comments. Notably, additional experiments argue against the study's original conclusion that FtsZ treadmilling and PG synthesis are fully independent in *B. subtilis*. Instead treadmilling appears to be important for efficient cell wall synthesis. Complete inhibition of FtsZ treadmilling reduces septal constriction rates up to ~30% depending on growth conditions. The relationship between FtsZ treadmilling and PG synthesis in *B. subtilis* thus remains unclear, albeit different from that in *S. aureus* and *S. pneumoniae* despite the relatively close evolutionary relationship between the three organisms. Altogether, the quality of the experiments, clarification that FtsZ treadmilling and PG synthesis are asynchronous, and data suggesting treadmilling might be important for PG synthesis efficiency, make this a very useful study which will influence future work in the field.

Reviewer #2:

Remarks to the Author:

I read thoroughly each of the previous comments and responses and the revised version of this paper. The authors did an outstanding job in addressing each previous comment completely, in many cases by adding new data or calculating new parameters from existing data. The changes and additional new data add substantially to this revision, which is more convincing, informative, and complete than the first version. In particular, the authors addressed all of the critical points raised in the previous review by the reviewers and editors. They included important new data about the numbers of molecules moving in different motions. They included additional controls, as requested. They expanded their analysis of the temperature dependence of FtsZ treadmilling and divisome complex multimerization. These additions led to refined major conclusions compared the first version. For example, the revised version concludes that there is an indirect effect of FtsZ treadmilling on PBP2B speed and that there is no direct sPG synthesis regulation by FtsZ treadmilling or involvement of disordered, fragmented FtsZ tracks.

This revision is highly informative, original, and rigorous. It is well presented and convincing. This paper is an important contribution that will have considerable impact on the field.

I have only two minor points.

Scientific:

Line 106. I suggest the following change for clarity: "immobile PBP2B molecules are bound noncovalently to acceptor peptides". ("or interact noncovalently with acceptor stems")

4

The reason for this clarification is that PBP2B also binds the acceptor peptide covalently in the transpeptidase reaction, which occurs after the glycosyltransferase reaction in active sPG synthesis. The binding mentioned here based on reference 18 is independent of catalysis.

Style:

Line 173. "FtsZ-associated speed population" seems awkward.

Reviewer #3:

Remarks to the Author:

The revised manuscript by Whitley et al. analyzes the dynamics of peptidoglycan synthesis during septation in *Bacillus subtilis*. The authors have done a tremendous amount of work responding to the suggestions of all the reviewers. I especially appreciated the new analysis of treadmilling at 21°C. The idea that treadmilling speed is regulated in vivo by other factors and not only GTPase activity is intriguing. Perhaps this factor(s) could be differentially expressed at lower temperatures? I have only some relatively minor suggestions to which the authors may or may not wish to respond.

Minor comments:

1. Fig. 2A. Please formally report the average speed of processive PBP2B (in the main text with errors, for example) since it is difficult to decipher this directly from the graphs.
2. Fig. 2B-C. It is difficult to distinguish between the light blue and green symbols in the legend. Consider making the symbols in the legend slightly larger.
3. Legend for Fig. S14. Please indicate that green corresponds to BADA and magenta corresponds to TADA.
4. The fact that HT-FtsW shows the same intensity drops as HT-PBP2B is consistent with the authors' hypothesis that the PG synthesis machinery is multimeric. To make this part of the work a bit more consistent, consider quantifying the percentage of tracks where they see intensity changes for FtsW (similar to what they reported for PBP2B).

Reviewer #4:

None

Reviewer #5:

None

Author Rebuttal, first revision:

Author response:

5We would like to thank all the reviewers for their positive words and helpful suggestions. We have revised the manuscript to address the remaining concerns raised by the reviewers.

Below we respond to the comments raised by the reviewers point by point.

We have noted in **bold** changes made to the manuscript.

Reviewer #1 (Remarks to the Author):

The authors have done an excellent job responding to the reviewers' comments. Notably, additional experiments argue against the study's original conclusion that FtsZ treadmilling and PG synthesis are fully independent in *B. subtilis*. Instead treadmilling appears to be important for efficient cell wall synthesis. Complete inhibition of FtsZ treadmilling reduces septal constriction rates up to ~30% depending on growth conditions. The relationship between FtsZ treadmilling and PG synthesis in *B. subtilis* thus remains unclear, albeit different from that in *S. aureus* and *S. pneumoniae* despite the relatively close evolutionary relationship between the three organisms. Altogether, the quality of the experiments, clarification that FtsZ treadmilling and PG synthesis are asynchronous, and data suggesting treadmilling might be important for PG synthesis efficiency, make this a very useful study which will influence future work in the field.

We thank the Reviewer for their positive comments.

Reviewer #2 (Remarks to the Author):

I read thoroughly each of the previous comments and responses and the revised version of this paper. The authors did an outstanding job in addressing each previous comment completely, in many cases by adding new data or calculating new parameters from existing data. The changes and additional new data add substantially to this revision, which is more convincing, informative, and complete than the first version. In particular, the authors addressed all of the critical points raised in the previous review by the reviewers and editors. They included important new data about the numbers of molecules moving in different motions. They included additional controls, as requested. They expanded their analysis of the temperature dependence of FtsZ treadmilling and divisome complex multimerization. These additions led to refined major conclusions compared the first version. For example, the revised version concludes that there is an indirect effect of FtsZ treadmilling on PBP2B speed and that there is no direct sPG synthesis regulation by FtsZ treadmilling or involvement of disordered, fragmented FtsZ tracks.

This revision is highly informative, original, and rigorous. It is well presented and convincing. This paper is an important contribution that will have considerable impact on the field.

We thank the Reviewer for their positive comments. We have now made changes to the text to address the Reviewer's remaining suggestions, detailed below.

I have only two minor points.

Scientific:

Line 106. I suggest the following change for clarity: "immobile PBP2B molecules are bound noncovalently to acceptor peptides". ("or interact noncovalently with acceptor stems")

The reason for this clarification is that PBP2B also binds the acceptor peptide covalently in the transpeptidase reaction, which occurs after the glycosyltransferase reaction in active sPG synthesis. The binding mentioned here based on reference 18 is independent of catalysis.

We have made this change.

Style:

Line 173. "FtsZ-associated speed population" seems awkward.

We have reworded this to "...a motile population of HT-PBP2B associated with FtsZ treadmilling..." (Line 155 of revised text).

Reviewer #3 (Remarks to the Author):

The revised manuscript by Whitley et al. analyzes the dynamics of peptidoglycan synthesis during septation in *Bacillus subtilis*. The authors have done a tremendous amount of work responding to the suggestions of all the reviewers. I especially appreciated the new analysis of treadmilling at 21oC. The idea that treadmilling speed is regulated in vivo by other factors and not only GTPase activity is intriguing. Perhaps this factor(s) could be differentially expressed at lower temperatures? I have only some relatively minor suggestions to which the authors may or may not wish to respond.

We thank the Reviewer for their positive comments. We have now made changes to the text to address the Reviewer's remaining suggestions, detailed below.

Minor comments:

1. Fig. 2A. Please formally report the average speed of processive PBP2B (in the main text with errors, for example) since it is difficult to decipher this directly from the graphs.

We have made a Supplementary Table with this information (Supplementary Table 2) and referenced this in the text.

2. Fig. 2B-C. It is difficult to distinguish between the light blue and green symbols in the legend. Consider making the symbols in the legend slightly larger.

We have made the symbols larger.

3. Legend for Fig. S14. Please indicate that green corresponds to BADA and magenta corresponds to TADA.

We have added this to the legend of Extended Data Figure 5 (formerly Supplementary Figure 14).

4. The fact that HT-FtsW shows the same intensity drops as HT-PBP2B is consistent with the authors' hypothesis that the PG synthesis machinery is multimeric. To make this part of the work a bit more consistent, consider quantifying the percentage of tracks where they see intensity changes for FtsW (similar to what they reported for PBP2B).

Quantification of the fluorescence drops and jumps of HT-FtsW will not provide meaningful information on the relative stoichiometries of FtsW and PBP2B due to substantial experimental differences between the data sets. The strains expressing the two fusion proteins used different promoters (HT-FtsW: P_{xyI} ; HT-PBP2B: $P_{\text{hyperspank}}$) and hence different inducers (HT-FtsW: 1% xylose; HT-PBP2B: 100 μM IPTG), meaning the levels of each protein in the imaged cells will not be directly comparable. Additionally, the degree of labelling was different between the two experiments (HT-FtsW: 5 nM JFX554 HaloTag ligand; HT-PBP2B: 100-250 pM JFX554 HaloTag ligand). The differing expression levels and degree of labelling between experiments mean that any difference in percentage of intensity drops / jumps would reflect differences in experimental parameters rather than biologically meaningful information on the relative stoichiometries of FtsW and PBP2B. We have therefore not quantified the HT-FtsW intensity drops or jumps in our data. However, we agree this is an important question. In future work we would very much like to quantify stoichiometry of different divisome complex components, likely using single molecule photobleaching approaches (Tashev et al, bioRxiv 2023).

Final Decision Letter:

Message: 23rd February 2024

Dear Dr Whitley,

I am pleased to accept your Article "Peptidoglycan synthesis drives a single population of septal cell wall synthases during division in *Bacillus subtilis*" for publication in Nature Microbiology. Thank you for having chosen to submit your work to us and many congratulations.

You may wish to make your media relations office aware of your accepted publication, in case they consider it appropriate to organize some internal or external publicity. Once your paper has been scheduled you will receive an email confirming the publication details. This is normally 3-4 working days in advance of publication. If you need additional notice of the date and time of publication, please let the production team know when you receive the proof of your article to ensure there is sufficient time to coordinate. Further information on our embargo policies can be found here:

<https://www.nature.com/authors/policies/embargo.html>

9Acceptance of your manuscript is conditional on all authors' agreement with our publication policies (see <https://www.nature.com/nmicrobiol/editorial-policies>). In particular your manuscript must not be published elsewhere.

Please note that *Nature Microbiology* is a Transformative Journal (TJ). Authors may publish their research with us through the traditional subscription access route or make their paper immediately open access through payment of an article-processing charge (APC). Authors will not be required to make a final decision about access to their article until it has been accepted. Find out more about Transformative Journals

With kind regards,